# Stability and Generalization of Split Learning: Sequential and Federated

## Abstract

Split Learning (SL) has emerged as a practical paradigm for training large models under privacy and systems constraints, showing strong performance on heterogeneous data and aligning well with LLM-era workloads. However, while convergence analyses for SL algorithms such as Sequential Split Learning (SSL) and Split Federated Learning (SFL) are well-established, their generalization bounds, especially those dependent on iteration-specific factors, remain largely unexplored, hindered by challenges like client drift and biased gradient estimates. In this work, we introduce the first theoretical framework for analyzing the generalization error of SL algorithms, leveraging an on-average stability approach to account for both local update drift and aggregation-induced errors. Our framework provides a novel connection between optimization and generalization, revealing how SSL and SFL differ in their stability profiles and generalization behavior. Specifically, we demonstrate that SSL excels in sparse client participation and long-horizon training, while SFL benefits from balanced participation in non-convex regimes, offering a clear guide for selecting the appropriate aggregation strategy. By deriving precise stability bounds for both convex and non-convex settings, we provide deep insights into the role of data heterogeneity, client drift, and aggregation mechanisms in SL. Extensive experiments on MNIST and CIFAR-10 benchmarks validate our theoretical predictions, highlighting the robustness and applicability of our framework across a range of practical scenarios.

## 1 Introduction

Pipeline parallelism is a key strategy for scaling large models, enabling efficient training and fine-tuning on edge devices. Split Learning (SL), a fundamental approach, partitions a neural network between clients and a central server, exchanging intermediate activations and gradients instead of raw data. This design mitigates data heterogeneity by enabling collaborative training across diverse data distributions without sharing raw data, while inherently enhancing privacy protection. SL is thus well-suited for privacy-sensitive applications like large language models (LLMs) (Zhao et al., 2024; Zhang et al., 2025b; He et al., 2025). As illustrated in 1, SL has inspired frameworks like Sequential Split Learning (SSL) (Gupta & Raskar, 2018) and Split Federated Learning (SFL) (Thapa et al., 2022) for deploying LLMs on edge devices with privacy-preserving features. In SSL, the model is sequentially passed among clients, with aggregation after each update. Conversely, SFL trains client-server splits in parallel, aggregating results per round, which enhances parallelism and accelerates convergence. These aggregation strategies provide distinct scalability and performance trade-offs based on application and participation constraints.

Despite significant advances in Split Learning (SL), the generalization properties—especially iterate-dependent bounds—have received limited study. While convergence analyses for Sequential Split Learning (SSL) and Split Federated Learning (SFL) are well established (Li & Lyu, 2023; Han et al., 2024), deriving iterate-dependent guarantees remains difficult due to SL-specific factors. First, data heterogeneity combined with partial model splits and cut layers yields biased gradients relative to end-to-end training. Second, SSL's sequential aggregation and SFL's parallel aggregation introduce randomness and client drift, altering optimization stability. Hence, precise measurement of client drift i.e., discrepancies in client updates induced by these aggregation schemes , is required. These issues motivate a tailored, algorithm-dependent stability analysis for SL.

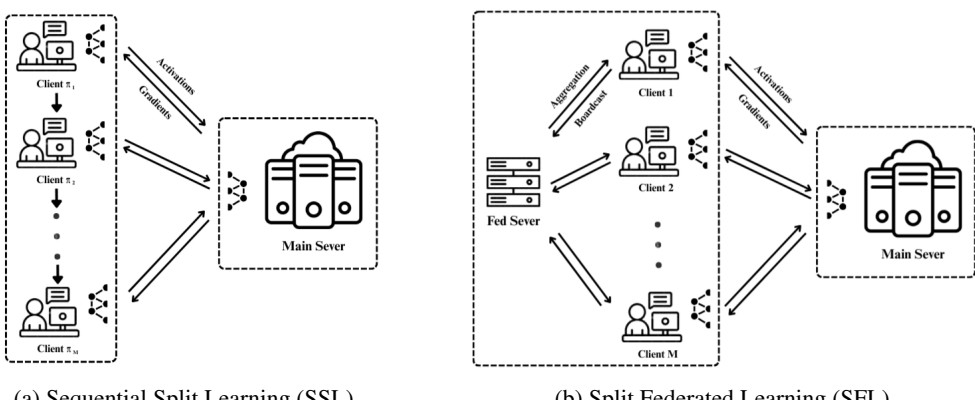

(a) Sequential Split Learning (SSL).          (b) Split Federated Learning (SFL).

Figure 1: Overviews of Split Learning paradigms.

In this paper, we propose an algorithm-dependent, on-average model stability framework (Lei & Ying, 2020) that quantifies a model's stability with single-sample perturbations. Under mild assumptions and two common step-size schedules, our method mirrors Split Learning's gradient flow, embedding per-round stability analysis. Specifically, we quantify (i) local update drift at each client and (ii) aggregation effects in Sequential Split Learning (SSL) and Split Federated Learning (SFL), then combine them to assess stability over training for both convex and smooth non-convex settings.

Our results yield key insights: in the non-convex case, both SSL and SFL achieve a clean $1/(MN)$ scaling with sublinear dependence on $TK$. In SSL, correction terms depend on the number of active clients, and the method remains robust under sparse or bursty participation. By contrast, SFL performs better when participation is moderate and stable across clients. Harmonic step-size schedules also suppress the exponential transients associated with square-root decay.

**Our Contributions.**

- *First work about stability and generalization in split learning.* We develop an algorithm-dependent stability calculus without $L$-Lipschitz loss assumptions, linking optimization to generalization. For harmonic and square-root learning rate decays, our bounds clarify how $\beta$, $T$, $K$, $M$, $N$, heterogeneity $\zeta$, and participation $q$, $\tilde{M}$ affect generalization (Table 1).

- *Choosing SSL or SFL.* We analyze SSL (sequential) and SFL (parallel) aggregation, showing their impact on stability and generalization. SSL suits sparse availability (low $\tilde{M}$) and convex objectives with large $T$. SFL excels in heterogeneous, non-convex systems with high client participation.

- *Experimental Validation.* We provide numerical experiments on MNIST with logistic regression on convex case and on CIFAR-10 with ResNet-18 on non-convex case to validate our theory. The preliminary results are consistent with our theoretical insights.

Table 1: Summary of Assumptions, Stability Tools, and Dependencies for Various Algorithms. LL: $L$-Lipschitz loss (or bounded gradients); SM: $\beta$-smooth; $M$: number of clients; $N$: number of samples per client; $\tilde{M}$: active clients per round (SSL); $q$: participation rate (SFL); $T$: number of rounds; $K$: local steps; $\eta$: stepsize; $\sigma$: stochastic noise; $\zeta$: heterogeneity.

| Algorithm | LL | SM | Stability Tool | $M$ | $N$ | $\tilde{M}/qT$ | $K$ | $\eta$ | $\sigma$ | $\zeta$ |
|---|---|---|---|---|---|---|---|---|---|---|
| FedAvg Sun et al. (2023) | ✗ | ✓ | Uniform stability | ✓ | ✓ | ✓ | ✓ | ✓ | ✓ | ✗ |
| FedAvg Sun et al. (2024) | ✓ | ✓ | On-average stability ($\ell_1$) | ✓ | ✓ | ✗ | ✓ | ✗ | ✓ | ✓ |
| D-SGD Bellet et al. (2024) | ✓ | ✓ | On-average stability ($\ell_1$) | ✓ | ✓ | ✗ | ✓ | - | ✓ | ✗ | ✗ |
| D-SGD Ye et al. (2025a) | ✓ | ✓ | On-average stability ($\ell_1$) | ✓ | ✓ | ✗ | ✓ | - | ✓ | ✓ | ✓ |
| **SSL/SFL (Ours)** | ✗ | ✓ | On-average stability ($\ell_2$) | ✓ | ✓ | ✓ | ✓ | ✓ | ✓ | ✓ |

## 2 RELATED WORK

**Split Learning.** Split Learning (SL) partitions a network between clients and a server, exchanging intermediate activations and gradients to preserve data privacy. Early work showed feasibility in multi-agent and healthcare settings (Gupta & Raskar, 2018; Vepakomma et al., 2018), and later comparisons with FL quantified communication and systems trade-offs. Two paradigms dominate: SSL and SFL (Gupta & Raskar, 2018; Thapa et al., 2022). SSL forwards partially updated models across clients, ensuring strict privacy and low per-round communication but incurring serialization latency (Gupta & Raskar, 2018); SSL research established convergence under heterogeneity and exposed privacy risks, prompting defenses against inference, label exposure, and model inversion (Li & Lyu, 2023; Pasquini et al., 2021; Li et al., 2021; Titcombe et al., 2021; Erdogan et al., 2022); systems reduced cost via server-side gradient averaging and learning-rate acceleration (Pal et al., 2021). SFL integrates FL-style parallelism with SL partitioning, enabling simultaneous client updates followed by aggregation (Thapa et al., 2022; McMahan et al., 2017); it is effective in constrained and wireless environments and admits convergence under heterogeneous, non-convex settings (Wu et al., 2023; Lin et al., 2024; Han et al., 2024). Applications include large-scale vision pretraining, secure learning under label-inference threats, and distributed LLM training via FedSLLM, federated splitting, and VFLAIR-LLM (Wang et al., 2023; Liu et al., 2024; Gao & Zhang, 2023; Zhao et al., 2024; Zhang et al., 2025b; Gu et al., 2025); recent systems—Hourglass, Ampere, Protocol Models—refine SFL with communication-efficient scheduling and model parallelism, improving scalability and accuracy (He et al., 2025; Zhang et al., 2025a; Ramasinghe et al., 2025).

**Stability and Generalization.** Algorithmic stability has long been a critical avenue for understanding generalization. Bousquet & Elisseeff (2002); Elisseeff et al. (2005) formalized uniform stability, and Hardt et al. (2016) proved that stochastic gradient descent (SGD) is uniformly stable, leading to extensions in non-convex settings, such as SGLD (Mou et al., 2018). Recent studies have applied uniform stability to analyze stability in Federated Learning (FL) and decentralized Stochastic Gradient Descent (D-SGD) (Sun et al., 2023; Wang et al., 2024; Liu et al., 2025), as well as stochastic weight averaging (Wang et al., 2024) and federated systems with partial client participation (Zhang et al., 2024). On-average stability, introduced by Lei & Ying (2020); Lei et al. (2023), has been extended to higher generalization bounds of D-SGD (Bellet et al., 2024) and complex settings such as heterogeneous federated learning and Byzantine-resilient D-SGD (Sun et al., 2024; Ye et al., 2025a;b). Our work builds on $\ell_2$ on-average stability, which alleviates the reliance on the Lipschitz condition, providing a more flexible tool for large-scale federated learning systems.

## 3 PRELIMINARIES

In this section, we present some notation and define the optimization and generakization in Sec 3.1, describe the SL update rules in Sec 3.2. For a detailed nomenclature, please refer to Appendix A.

### 3.1 PROBLEM SETUP

We consider a set of clients $\mathcal{M} = \{1, 2, \ldots, M\}$, each holding a private local distribution $\mathcal{D}_m$ supported on $\mathcal{Z}$. The objective of Split Learning is to learn a global model, parameterized by $\boldsymbol{\omega}$, that minimizes the *population risk*, as defined by:

$$\min_{\boldsymbol{\omega}} R(\boldsymbol{\omega}) \triangleq \sum_{m=1}^{M} c_m \, \mathbb{E}_{Z \sim \mathcal{D}_m} [f(\boldsymbol{\omega}; Z)], \qquad \boldsymbol{\omega}^* = \arg\min_{\boldsymbol{\omega}} R(\boldsymbol{\omega}).$$

Here, $f$ denotes the loss function, and $c_m \in (0, 1]$ is a weight proportional to the dataset size of client $m$ (typically, $c_m = \frac{|\mathcal{D}_m|}{\sum_{k=1}^{M} |\mathcal{D}_k|}$), such that $\sum_{m=1}^{M} c_m = 1$.

In practice, we solve this problem by a random algorithm $\mathcal{A}$ over empirical counterpart, computed over $M$ local datasets $S \triangleq (S_1, \ldots, S_M)$, where $S_m = \{Z_{1m}, \ldots, Z_{Nm}\}$ denotes the dataset of client $m$ with $Z_{im} \sim \mathcal{D}_m$. For simplicity, we assume all local datasets have the same size $N$, although our analysis can be extended to accommodate heterogeneous sizes. The corresponding

*empirical risk minimization* problem is formulated as:

$$\min_{\boldsymbol{\omega}} R_S(\boldsymbol{\omega}) \triangleq \sum_{m=1}^{M} c_m\, R_{S_m}(\boldsymbol{\omega}) \triangleq \frac{1}{N} \sum_{m=1}^{M} \sum_{n=1}^{N} c_m\, f(\boldsymbol{\omega}; Z_{nm}), \qquad \boldsymbol{\omega}_S^* = \arg\min_{\boldsymbol{\omega}} R_S(\boldsymbol{\omega}).$$

**Definition 3.1.** Given a dataset $S$ and a randomized algorithm as a map $\mathcal{A}: S \to \boldsymbol{\Omega}$, we define:

- *Generalization error* is defined as $\epsilon_{\text{gen}} = \mathbb{E}_{S,A}[R(\mathcal{A}(S)) - R_S(\mathcal{A}(S))]$, i.e., the expected statistical discrepancy between the population and empirical risk distributions.

- *Excess generalization error* is defined as $\epsilon_{\text{exc}} = \mathbb{E}_{S,A}[R(\mathcal{A}(S)) - R(\omega^*)]$, i.e., the expected performance gap between the population risk and the global true minimizer.

- *Optimization error* is defined as $\epsilon_{\text{opt}} = \mathbb{E}_{S,A}[R_S(\mathcal{A}(S)) - R_S(\omega_S^*)]$, i.e., the expected convergence gap between the population risk and the empirical risk minimizer solution.

Furthermore, the excess generalization error $\epsilon_{\text{exc}}$ can be decomposed as follows:

$$\epsilon_{\text{exc}} = \underbrace{\mathbb{E}_{S,\mathcal{A}}\left[R(\mathcal{A}(S)) - R_S(\mathcal{A}(S))\right]}_{\epsilon_{\text{gen}}} + \underbrace{\mathbb{E}_{S,\mathcal{A}}\left[R_S(\mathcal{A}(S)) - R_S(\boldsymbol{\omega}_S^*)\right]}_{\epsilon_{\text{opt}}} + \underbrace{\mathbb{E}_{S,\mathcal{A}}\left[R_S(\boldsymbol{\omega}_S^*) - R(\boldsymbol{\omega}^*)\right]}_{\leq 0}.$$

Thus, the excess generalization error combines optimization error and generalization error together.

## 3.2 ALGORITHM

Split Learning (SL) works by dividing a deep model into two parts at a designated cut layer, $L_c$: the client-side model, consisting of the first $L_c$ layers, and the server-side model, consisting of the remaining layers. The **training process** consists of the following steps:

**Client Forward Propagation.** Each client performs a forward pass on its local model to compute the smashed data (i.e., the activations at the cut layer). These activations, along with the corresponding labels, are transmitted to the server. All clients operate in parallel during this step.

**Server-Side Training.** After receiving the smashed data, the server proceeds with the forward pass on its portion of the model, computes the loss, and updates the server-side parameters. It also computes the gradient respect to the cut-layer activations and sends this gradient back to the clients.

**Client Backward Propagation.** Each client receives the gradient signal from the server and uses the chain rule to complete backpropagation on its local parameters, updating the client-side model.

After several steps of local training, model aggregation or parameter passing is performed among the involved participants, ensuring consistent global model updates.

Two representative variants of this framework are commonly considered: (i) *Sequential Split Learning (SSL)* Gupta & Raskar (2018), in which clients train sequentially by passing the updated parameters from one to the next; and (ii) *Split Federated Learning (SFL)* Thapa et al. (2022), in which clients train in parallel and the global model is aggregated using federated averaging. We now describe their aggregation mechanisms and update rules in detail.

### 3.2.1 SEQUENTIAL SPLIT LEARNING (SSL)

We present a concise version of SSL in Algorithm 1 to illustrate its update rules. At the start of each training round, indices $\pi_1, \pi_2, \ldots, \pi_{\tilde{M}}$ are randomly sampled without replacement from $\{1, 2, \ldots, M\}$, forming a random permutation that determines the $\tilde{M}$ involved clients' training order. In each round, the first client, $\pi_1$, receives the current global parameter vector and performs $K$ steps of local updates using its local dataset. The updated local parameter is passed to the next client, and this process continues until all clients complete their local training. Let $\boldsymbol{\omega}_m^{(t,k)}$ denote the local parameter of client $\pi_m$ after $k$ updates in round $t$, and let $\boldsymbol{\omega}^{(t)}$ and $g_m^{(t,k)}$ represents the the model parameter and the gradient of the loss function with respect to it. Using stochastic gradient descent (SGD) as the local solver, the SSL update rule is as follows:

$$\text{Local update:} \quad \boldsymbol{\omega}_m^{(t,k+1)} = \boldsymbol{\omega}_m^{(t,k)} - \eta_m^{(t,k)} g_{\pi_m}^{(t,k)}, \quad \text{with initial} \quad \boldsymbol{\omega}_m^{(t,0)} = \begin{cases} \boldsymbol{\omega}^{(t)}, & \text{if } m = 1, \\ \boldsymbol{\omega}_{m-1}^{(t,K)}, & \text{if } m > 1. \end{cases}$$

Global aggregation: $\quad \boldsymbol{\omega}^{(t+1)} = \boldsymbol{\omega}_{\pi_M}^{(t,K)}$.

### 3.2.2 Split Federated Learning (SFL)

In contrast to SSL, Split Federated Learning (SFL) shown in Algorithm 2 allows clients to train in parallel. At the start of each communication round, every client downloads the latest global client-side parameters. The clients then perform local training, interacting with the server through forward and backward propagation, as described in the general SL framework. After $K$ local epochs, the server aggregates the client-side models using a weighted averaging scheme (e.g., FedAvg McMahan et al. (2017)) to form the updated global client-side parameters. Partial participation is controlled by the set $\tilde{\mathcal{M}}_{\sqcup}$, which includes the indices of the clients participating in round $t$. The server-side models are also aggregated across clients to update the global server-side parameters. The update rules for SFL are summarized as follows:

Local update: $\quad \boldsymbol{\omega}_m^{(t,k+1)} \leftarrow \boldsymbol{\omega}_m^{(t,k)} - \eta^{(t,k)} g_m^{(t,k)}$,

Global aggregation: $\boldsymbol{\omega}^{(t+1)} \leftarrow \boldsymbol{\omega}^{(t)} - \sum_{m \in \tilde{\mathcal{M}}_{\sqcup}} c_m \sum_{k=0}^{K} \eta^{(t,k)} g_m^{(t,k)}$, with $\quad \sum_{m \in \tilde{\mathcal{M}}_{\sqcup}} c_m = q \leq 1$ or more details about the two algorithms and their pseudocode, please refer to Appendix C.

## 4 Theoretical Analysis

In this section, we provide the necessary assumptions in Sec 4.1, then study on-average model stability bound and excess generalization bound of SSL and SFL in Sec 4.2. The discussion about discovery and insight of theorems is contained in Sec 4.3.

### 4.1 Assumption

**Assumption 4.1** ($\beta$-smoothness). The loss function $f$ is $\beta$-smooth, i.e., there exists $\beta > 0$ such that for all $\boldsymbol{\omega}, \boldsymbol{\omega}' \in \mathbb{R}^d, z \in \mathcal{Z}$,

$$\|\nabla f(\boldsymbol{\omega}; z) - \nabla f(\boldsymbol{\omega}'; z)\|_2 \leq \beta \|\boldsymbol{\omega} - \boldsymbol{\omega}'\|_2.$$

**Assumption 4.2** (Bounded Stochastic Gradient Variance). The stochastic gradients at each client are unbiased, and their variance is bounded by $\sigma^2$:

$$\mathbb{E}_{Z_{mn}} \|\nabla f(\boldsymbol{\omega}; Z_{mn}) - \nabla R_{S_m}(\boldsymbol{\omega})\|^2 \leq \sigma^2,$$

for any agent $m \in \mathcal{M}$ and $\boldsymbol{\omega} \in \mathbb{R}^d$.

**Assumption 4.3** (Bounded Heterogeneity). There exists $\zeta^2 > 0$ such that for any $\boldsymbol{\zeta} \in \mathbb{R}^d$,

$$\frac{1}{M} \sum_{m=1}^{M} \|\nabla R_{S_m}(\boldsymbol{\omega}) - \nabla R_S(\boldsymbol{\omega})\|^2 \leq \zeta^2,$$

*Remark* 4.1. The smoothness assumption is common uesd in stability analysis (Lei & Ying, 2020; Sun et al., 2024; Bellet et al., 2024), and it is valid for many loss functions, such as logistic regression, softmax classifiers, and $l_2$-norm regularized linear regression. The value of $\sigma$ quantifies the level of stochasticity, while a larger value of $\zeta^2$ indicates a degree of data heterogeneity.

### 4.2 Stability and Generalization

First, we give the definition of on-average model stability as follows:

**Definition 4.1** ($\ell_2$ On-average Model Stability). (Lei & Ying, 2020) Let $S = (S_1, \ldots, S_M)$ with $S_m = \{Z_{1m}, \ldots, Z_{Nm}\}$ and $\tilde{S} = (\tilde{S}_1, \ldots, \tilde{S}_M)$ with $\tilde{S}_m = \{\tilde{Z}_{1m}, \ldots, \tilde{Z}_{Nm}\}$ be two independent copies such that $Z_{im} \sim \mathcal{D}_m$ and $\tilde{Z}_{im} \sim \mathcal{D}_m$. For any $i \in \{1, \ldots, N\}$ and $j \in \{1, \ldots, M\}$, we define $S^{(ij)} = (S_1, \ldots, S_{j-1}, S_j^{(i)}, S_{j+1}, \ldots, S_M)$, where $S_j^{(i)} = \{Z_{1j}, \ldots, Z_{i-1j}, \tilde{Z}_{ij}, Z_{i+1j}, \ldots, Z_{Nj}\}$ is the dataset formed from $S$ by replacing the $i$-th element of the $j$-th agent's dataset with $\tilde{Z}_{ij}$. Algorithm $\mathcal{A}$ is said to be $l_2$ *on-average model $\varepsilon$-stable* if

$$\mathbb{E}_{S, \tilde{S}, \mathcal{A}} \left[ \frac{1}{MN} \sum_{i=1}^{N} \sum_{j=1}^{M} \|\mathcal{A}(S) - \mathcal{A}(S^{(ij)})\|_2^2 \right] \leq \varepsilon^2.$$

*Remark* 4.2. The definition of $\ell_2$ on-average model stability indicates how robust the algorithm is to perturbations in the local datasets, which quantifies the sensitivity of the output model to the replacement of individual data points in the datasets.

**Theorem 4.1** (Generalization via on-average model stability). *(Lei & Ying, 2020) Let $\mathcal{A}$ be an $\ell_2$ on-average model $\varepsilon$-stable algorithm. If the convex loss function $f(\cdot; z)$ is nonnegative and $\beta$-smooth for all $z \in \mathcal{Z}$, we have the generalization bound with constant $\gamma > 0$:*

$$\epsilon_{gen} \leq \frac{1}{2MN\gamma} \sum_{i=1}^{N} \sum_{j=1}^{M} \mathbb{E}_{S,\mathcal{A}} \left[ \|\nabla f(\mathcal{A}(S); Z_{ij})\|^2 \right] + \frac{\beta + \gamma}{2MN} \sum_{i=1}^{N} \sum_{j=1}^{M} \mathbb{E}_{S,\tilde{S},\mathcal{A}} \left[ \|\mathcal{A}(S) - \mathcal{A}(S^{(ij)})\|^2 \right]$$

**Theorem 4.2** (Generalization via on-average model stability). *Assume that the loss function $f(\cdot, z)$ is nonnegative and bounded in $[0, 1]$, and that $\beta$-smoothness holds. For all $i = 1, \ldots, N$ and $j = 1, \ldots, M$, let $\{\boldsymbol{\omega}^{(t)}\}_{t=0}^{T}$ and $\{\tilde{\boldsymbol{\omega}}^{(t)}\}_{t=0}^{T}$ denote the iterates for algorithms run on $S$ and $S^{(ij)}$ respectively, with $\Delta_t = \|\boldsymbol{\omega}^{(t)} - \tilde{\boldsymbol{\omega}}^{(t)}\|_2^2$. Then, for every $t_0 \in \{0, 1, \ldots, T\}$, we have the following bound for the generalization error with constant $\gamma > 0$:*

$$\epsilon_{gen} \leq \frac{t_0}{MN} + \frac{1}{2MN\gamma} \sum_{i=1}^{N} \sum_{j=1}^{M} \mathbb{E}_{S,\mathcal{A}} \left[ \|\nabla f(\mathcal{A}(S); Z_{ij})\|^2 \right] + \frac{\beta + \gamma}{2MN} \sum_{i=1}^{N} \sum_{j=1}^{M} \mathbb{E}_{S,\tilde{S},\mathcal{A}} \left[ \|\mathcal{A}(S) - \mathcal{A}(S^{(ij)}) \mid \Delta^{(t_0)} = \mathbf{0}\|^2. \right]$$

*Remark* 4.3. These theorem provides a generalization bound based on the smoothness and stability of the algorithm. It suggests that the generalization error can be controlled by both the gradient bound of the loss function and the stability of the algorithm under perturbations in the data.

According to the theorem, it suffices to control the on-average model stability of the algorithm $\mathcal{A}$ to obtain the desired generalization bound. For each round $t$, we define basic block as below: $S_t \triangleq \sum_{k=0}^{K-1} \eta^{(t,k)}$, $H_t \triangleq \sum_{k=0}^{K-1} \left( \eta^{(t,k)} \right)^2$, $Q_t \triangleq \sum_{k=0}^{K-1} \left( \eta^{(t,k)} \right)^2 \sum_{s=0}^{k} \left( \eta^{(t,s)} \right)^2$, and let $A_\star \triangleq \sigma^2 + \zeta^2 + \sup_t \|\nabla R_S(\boldsymbol{\omega}^{(t)})\|_2^2$, then we develop theorem below:

**Theorem 4.3** (On-average model stability and generalization error for SSL in the convex case). *Under Assumptions 4.1–4.3, suppose that the loss function is convex, with step sizes $\{\eta^{(t,k)}\} \leq \frac{2}{\beta}$. Then the expected on-average model stability satisfies*

$$\frac{1}{MN} \sum_{i=1}^{N} \sum_{j=1}^{M} \mathbb{E}_{S,\tilde{S},A} \left[ \|A(S) - A(S^{(ij)})\|^2 \right] \leq \frac{16\,\tilde{M}\,A_\star}{MN\,T} \left( \sum_{t=0}^{T-1} H_t + 4\beta^2 \sum_{t=0}^{T-1} Q_t \right).$$

*(i) For square-root decaying step sizes $\eta^{(t,k)} = \frac{1}{\sqrt{tK+k+k_0}}$, with $k_0 > 1$:*

$$\epsilon_{\text{gen}} \leq 4\sqrt{3}\,A_\star \sqrt{\frac{\tilde{M}}{MNT}} \sqrt{\frac{1}{k_0 - 1} + \frac{\beta^2}{K(k_0 - 1)}} + \frac{8\beta\,\tilde{M}\,A_\star}{MNT} \left[ \frac{1}{k_0 - 1} + 2\beta^2 \left( \frac{1}{K(k_0 - 1)} + \frac{1}{3(k_0 - 1)^3} \right) \right].$$

*(ii) For harmonically decaying step sizes $\eta^{(t,k)} = \frac{1}{tK+k+k_0}$, with $k_0 > 1$:*

$$\epsilon_{\text{gen}} \leq 4\sqrt{3}\,A_\star \sqrt{\frac{\tilde{M}}{MNT}} \sqrt{\frac{1}{k_0 - 1} + \frac{\beta^2}{K(k_0 - 1)}} + \frac{8\beta\,\tilde{M}\,A_\star}{MNT} \left[ \frac{1}{k_0 - 1} + 2\beta^2 \left( \frac{1}{K(k_0 - 1)} + \frac{1}{3(k_0 - 1)^3} \right) \right].$$

*Proof.* See Appendix E.1.2 for the proof. $\qquad\square$

**Corollary 4.1** (Excess Generalization Error). *The preview work Li & Lyu (2023) provides an analysis of $\varepsilon_{\text{opt}}$. The convergence rate of SSL is dominated by $\mathcal{O}(1/\sqrt{\tilde{M}KT})$ when $\eta \leq \Theta(1/(MK))$. Therefore, the excess risk of SSL in the convex case satisfies $\varepsilon_{\text{exc}} = \mathcal{O}\left( \sqrt{\frac{\tilde{M}}{MNT}} \right) + \mathcal{O}\left( \frac{1}{\tilde{M}\,K\,T} \right)$.*

**Theorem 4.4** (On-average model stability and generalization error for SSL in the non-convex case). *Under Assumptions 4.1–4.3, suppose that the loss function is non-convex. Then the expected on-average model stability of the output satisfies*

$$\frac{1}{MN} \sum_{i=1}^{N} \sum_{j=1}^{M} \mathbb{E}_{S,S^{(ij)},A} \left[ \|A(S) - A(S^{(ij)})\|_2^2 \right] \leq \frac{16\,\tilde{M}\,A_\star}{MN\,(T - t_0)} \sum_{t=t_0}^{T-1} e^{2\beta S_t} \left( H_t + 4\beta^2 Q_t \right) + \frac{t_0}{MN}.$$

*(i) For square-root decaying step sizes $\eta^{(t,k)} = \frac{1}{\sqrt{tK+k+k_0}}$ with $k_0 > 1$:*

$$\epsilon_{\mathrm{gen}} \lesssim \frac{(TK)^{\frac{2\beta}{1+2\beta}}}{MN} + \frac{8\tilde{M}\,A_\star}{MN\,T}\,(TK)^{\frac{\beta}{1+2\beta}-\frac{1}{2}}\,e^{2\beta\,(TK)^{\frac{1-\beta}{2(1+2\beta)}}} + 4A_\star\sqrt{\frac{3\tilde{M}}{\beta MN\,T}}\,(TK)^{\frac{\beta}{2(1+2\beta)}-\frac{1}{4}}\,e^{\beta\,(TK)^{\frac{1-\beta}{2(1+2\beta)}}}.$$

*(ii) For harmonically decaying step sizes $\eta^{(t,k)} = \frac{1}{tK+k+k_0}$ with $k_0 > 1$:*

$$\epsilon_{\mathrm{gen}} \lesssim \frac{(TK)^{\frac{2\beta}{1+2\beta}}}{MN} + \frac{8\beta}{1+2\beta}\cdot\frac{\tilde{M}\,A_\star\,\log T}{MN\,T\,K} + 4A_\star\sqrt{\frac{3}{1+2\beta}}\sqrt{\frac{\tilde{M}\,\log T}{MN\,T\,K}}.$$

*Proof.* See Appendix E.1.3 for the proof. $\qquad\square$

**Corollary 4.2** (Excess Generalization Error). *According to the results in (Li & Lyu, 2023), let $\eta \leq \Theta\big(1/(MK)\big)$. Then the excess error is mainly goverment by:* $\varepsilon_{\mathrm{exc}} = \mathcal{O}\Big(\frac{(TK)^{\frac{2\beta}{1+2\beta}}}{MN}\Big) + \mathcal{O}\Big(\frac{1}{\sqrt{\tilde{M}KT}}\Big).$

**Theorem 4.5** (SFL On-Average Model Stability and Generalization in the Convex Case). *Under Assumptions 4.1–4.3, suppose the loss is convex and the step sizes satisfy $\{\eta^{(t,k)}\}_{k=0}^{K-1} \leq \frac{2}{\beta}$. Then the on-average model stability of the averaged output satisfies*

$$\frac{1}{MN}\sum_{i=1}^{N}\sum_{j=1}^{M}\mathbb{E}_{S,S^{(ij)},A}\left[\left\|A(S) - A\big(S^{(ij)}\big)\right\|^2\right] \leq \frac{16\,q\,A_\star}{MN(1-q)}\left(\sum_{t=0}^{T-1}H_t + 4\beta^2\sum_{t=0}^{T-1}Q_t\right).$$

*(i) For step sizes with square-root decay, $\eta^{(t,k)} = 1/\sqrt{tK+k+k_0}$ with $k_0 > 1$:*

$$\epsilon_{\mathrm{gen}} \leq 4\sqrt{3}\,A_\star\sqrt{\frac{q}{MN(1-q)}\left(\log\frac{TK+k_0-1}{k_0-1} + \frac{2\beta^2(K+1)}{k_0-1}\right)} + \frac{8\beta\,q\,A_\star}{MN(1-q)}\left(\log\frac{TK+k_0-1}{k_0-1} + \frac{2\beta^2(K+1)}{k_0-1}\right).$$

*(ii) For step sizes with harmonic decay, $\eta^{(t,k)} = 1/(tK+k+k_0)$ with $k_0 > 1$:*

$$\epsilon_{\mathrm{gen}} \leq 4\sqrt{3}\,A_\star\sqrt{\frac{q}{MN(1-q)}\left(\frac{1}{k_0-1} + \frac{2\beta^2}{K(k_0-1)} + \frac{2\beta^2}{3(k_0-1)^3}\right)} + \frac{8\beta\,q\,A_\star}{MN(1-q)}\left(\frac{1}{k_0-1} + \frac{2\beta^2}{K(k_0-1)} + \frac{2\beta^2}{3(k_0-1)^3}\right).$$

*Proof.* See Appendix E.2.2. $\qquad\square$

**Corollary 4.3** (Excess Generalization Error). *With the optimization results in (Han et al., 2024) as well, the excess generalization error of SFL in non-convex case satisfies with the $\varepsilon_{\mathrm{exc}} = \tilde{\mathcal{O}}\Big(\frac{q\log(TK)}{(1-q)MN}\Big) + \tilde{\mathcal{O}}\Big(\frac{M}{\sqrt{T}}\Big).$*

**Theorem 4.6** (SFL On-Average Model Stability and Generalization in the Non-Convex Case). *Under Assumptions 4.1–4.3, for any burn-in index $t_0 \in \{0, \ldots, T-1\}$, the on-average model stability of the output satisfies*

$$\frac{1}{MN}\sum_{i=1}^{N}\sum_{j=1}^{M}\mathbb{E}_{S,S^{(ij)},A}\left[\left\|A(S) - A\big(S^{(ij)}\big)\right\|^2\right] \leq \frac{16\,q\,A_\star}{MN(1-q)}\sum_{t=t_0}^{T-1}e^{2\beta S_t}\big(H_t + 4\beta^2 Q_t\big) + \frac{t_0}{MN}.$$

*(i) For step sizes with square-root decay, $\eta^{(t,k)} = 1/\sqrt{tK+k+k_0}$ with $k_0 > 1$:*

$$\epsilon_{\mathrm{gen}} \lesssim \frac{(TK)^{\frac{2\beta}{1+2\beta}}}{MN} + \frac{8\,q\,A_\star}{MN(1-q)}\,(TK)^{\frac{\beta}{1+2\beta}}\,K^{\frac{1}{2}}\,e^{2\beta\,(TK)^{\frac{1-\beta}{2(1+2\beta)}}}$$
$$+ 4\sqrt{3}\,A_\star\sqrt{\frac{q}{\beta MN(1-q)}}\,(TK)^{\frac{\beta}{2(1+2\beta)}}\,K^{\frac{1}{4}}\,e^{\beta\,(TK)^{\frac{1-\beta}{2(1+2\beta)}}}.$$

*(ii) For step sizes with harmonic decay, $\eta^{(t,k)} = 1/(tK+k+k_0)$ with $k_0 > 1$:*

$$\epsilon_{\mathrm{gen}} \lesssim \frac{(TK)^{\frac{2\beta}{1+2\beta}}}{MN} + \frac{2\beta\,A_\star}{(1+2\beta)\,MN}\cdot\frac{\log T}{TK} + \frac{2\sqrt{3}\,A_\star}{\sqrt{(1+2\beta)\,MN}}\cdot\sqrt{\frac{\log T}{TK}}.$$

*Proof.* See Appendix E.2.3. $\qquad\square$

**Corollary 4.4** (Excess Generalization Error). *Similarly, from the convergence result in the (Han et al., 2024), the excess generalization error satisfies $\varepsilon_{\mathrm{exc}} = \tilde{\mathcal{O}}\Big(\frac{(TK)^{\frac{2\beta}{1+2\beta}}}{MN}\Big) + \tilde{\mathcal{O}}\Big(\frac{1}{\sqrt[3]{T}}\Big).$*

### 4.3 DISCUSSION

Here we give some insight inside theorems and corollaries above :

*Remark* 4.4 (Influential Factors of the Generalization Error). Fixing model, loss, and dataset essentially makes $\beta$, $\sigma$, and $\zeta$ constants throughout. Theorem bounds clearly suggest: **(i)** enlarging per-client sample size $N$; **(ii)** increasing the number of clients $M$; **(iii)** reducing optimization distance to shrink $\sup_t \|\nabla R_S(\boldsymbol{\omega}^{(t)})\|_2^2$ in $A_\star$; **(iv)** using a smaller stepsize $\eta$ while still preserving convergence. Choosing more i.i.d. data further lowers $\zeta$ and thereby tightens the bound significantly.

*Remark* 4.5 (Stepsize Choice). In convex problems, square-root and harmonic decay yield quite similar leading-order bounds; thus square-root is usually simpler to tune effectively. In non-convex settings, square-root's cumulative step size causes $e^{2\beta S_t}$ to blow up rapidly, destabilizing the training, while harmonic decay keeps the stability term bounded, thereby improving robustness. Though square-root can sometimes speed early optimization (Li & Lyu, 2023; Han et al., 2024), harmonic decay preserves tighter generalization and successfully avoids exponential growth.

*Remark* 4.6 (Impact of Participation). In SSL, longer gradient paths naturally amplify output sensitivity: selecting $\tilde{M}$ clients per round adds gradients, so the bound usually grows with $\tilde{M}$. SFL averages gradients each round, maintaining better stability even with consistently high participation. For convex cases, bounds typically scale with $\mathcal{O}\left(\frac{q}{1-q}\right)$ ; smaller $q$ generally improves the overall generalization. In non-convex settings, participation effects drop down to lower-order terms.

*Remark* 4.7 (When to choose SSL or SFL). SSL's per-round averaging suits large $T$ and sparse edge devices; SFL benefits dense participation by aggregating many diverse gradients. For very large $T$ with non-convex objectives, both achieve essentially the same leading rate $\tilde{O}\big((TK)^{\frac{2\beta}{1+2\beta}}/(MN)\big)$; practical gaps stem mainly from subtle step-size and aggregation nuances.

Due to the limint of pages, more discussion about comparison to other generalization bounds in other multi agent algorithms (like FedAvg and D-SGD) is in Appendix C

## 5 EXPERIMENTAL RESULTS

In this Section, we validate our theory with classification experiments on logistic regression (Section 5.1) and ResNet (Section 5.2), and study how key factors affect stability errors.

### 5.1 LOGISTIC REGRESSION

In the validation of the convex objectives, we adopt classical logistic regression problem to validate the generalization in the training. We conduct experients on MINST dataset LeCun et al. (2002).

The experimental results in Figure 2 shows : **(i)** Square-root decay yields faster growth of instability $\|\omega_t - \omega_t'\|$, whereas harmonic decay converges more gently (Fig. 2(a)(d)), consistent with our theorem : slower decay suppresses cumulative perturbations and lowers $\epsilon_{\text{gen}}$. **(ii)** With a constant learning rate and fixed total iterations $TK$, increasing local updates $K$ markedly amplifies instability (Fig. 2(b)(e)), indicating larger client drift . **(iii)** Higher client participation consistently improves stability (Fig. 2(c)(f)), mitigating gradient variance and drift accumulation.

### 5.2 RESNET-18

We also conduct the experiments on ResNet-18 He et al. (2016) with CIFAR-10Krizhevsky et al. (2009) dataset to validate the properties in non-convex objectives.

The experimental results in Figure 3 shows : **(i)** Larger learning rates cause pronounced instability $\|\omega_t - \omega_t'\|$ in both SSL and SFL (Fig. 3(a)(d)), confirming our on-average $\ell_2$ stability analysis: slower decay or smaller steps better control parameter drift and reduce $\epsilon_{\text{gen}}$. **(ii)** Increasing the number of total client number markedly reduces instability and smooths the trajectories in SSL and SFL (Fig. 3(b)(e)), as averaging across more clients lowers gradient variance and mitigates the impact of heterogeneity $\zeta$. **(iii)** Higher client participation consistently improves stability (Fig. 3(c)(f)), mitigating gradient variance and drift accumulation. Overall, these non-convex results further val-

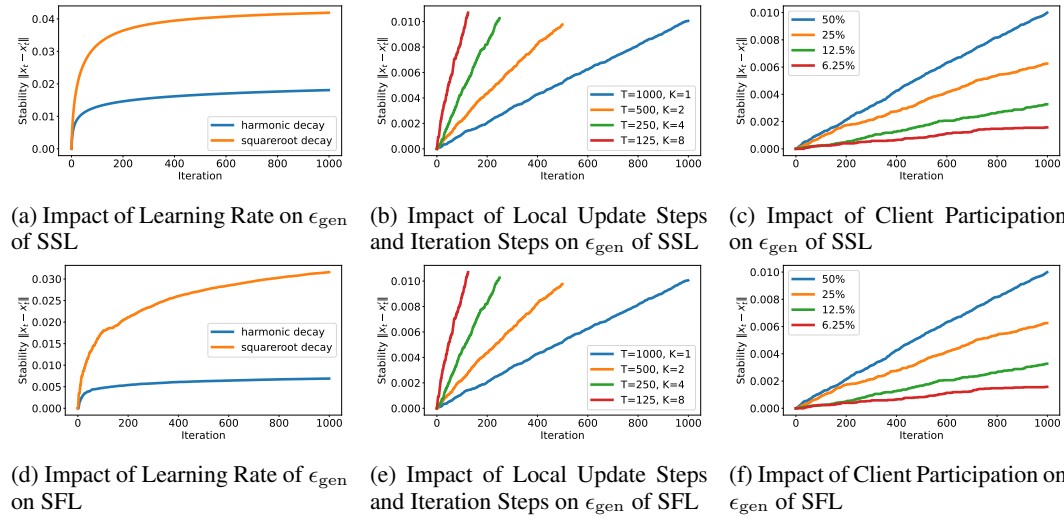

(a) Impact of Learning Rate on $\epsilon_{\text{gen}}$ of SSL

(b) Impact of Local Update Steps and Iteration Steps on $\epsilon_{\text{gen}}$ of SSL

(c) Impact of Client Participation on $\epsilon_{\text{gen}}$ of SSL

(d) Impact of Learning Rate of $\epsilon_{\text{gen}}$ on SFL

(e) Impact of Local Update Steps and Iteration Steps on $\epsilon_{\text{gen}}$ of SFL

(f) Impact of Client Participation on $\epsilon_{\text{gen}}$ of SFL

Figure 2: Generalization errors for a convex objective.

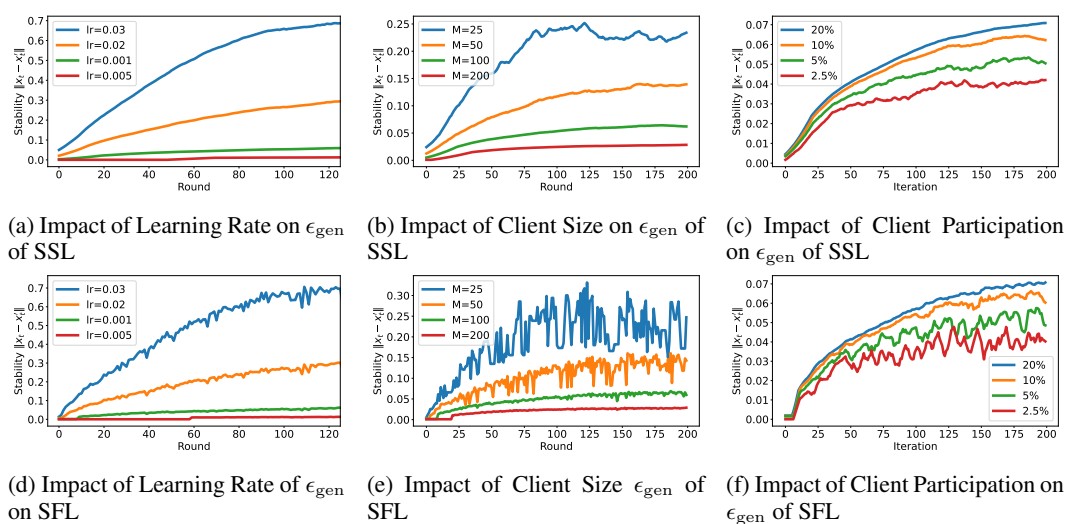

(a) Impact of Learning Rate on $\epsilon_{\text{gen}}$ of SSL

(b) Impact of Client Size on $\epsilon_{\text{gen}}$ of SSL

(c) Impact of Client Participation on $\epsilon_{\text{gen}}$ of SSL

(d) Impact of Learning Rate of $\epsilon_{\text{gen}}$ on SFL

(e) Impact of Client Size $\epsilon_{\text{gen}}$ of SFL

(f) Impact of Client Participation on $\epsilon_{\text{gen}}$ of SFL

Figure 3: Generalization errors for a non-convex objective.

idate our stability bounds and show the same key levers—moderate learning rate, controlled local updates, and broad participation—are essential for reducing $\epsilon_{\text{gen}}$ even beyond the convex case.

## 6 CONCLUSION

This paper provides the first comprehensive analysis of generalization error bounds for Split Learning (SL), focusing on Sequential Split Learning (SSL) and Split Federated Learning (SFL) in non-convex settings. Using an on-average stability framework, we quantify model responses to perturbations, offering generalization guarantees without assuming $L$-Lipschitz loss. Our findings highlight how client drift, aggregation schemes, and data heterogeneity affect stability and generalization, clarifying SSL and SFL behavior under different strategies. We show harmonic learning rate schedules mitigate transient effects of square-root decay, enhancing convergence in both convex and non-convex settings. Experiments on benchmark datasets validate our theoretical insights.

**Limitation.** The impact of cut layer placement on client drift and stability remains under-explored, with limited research on its convergence properties, which is a key direction for future work.

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

# APPENDIX

## A  APPENDIX: NOTATIONS

Table 2: Unified notation used throughout the paper.

| Symbol | Meaning |
|---|---|
| $\mathcal{M}$, $M$ | Set of clients and its size: $\mathcal{M} = \{1, \ldots, M\}$, $M = |\mathcal{M}|$. |
| $m$, $j$ | Client index ($m$ or $j \in \{1, \ldots, M\}$). |
| $N$, $i$ | Local sample count per client $N$; sample index $i \in \{1, \ldots, N\}$. |
| $D_m$, $\mathcal{Z}$ | Data distribution of client $m$ ($D_m$); sample space $\mathcal{Z}$. |
| $S$, $S_m$ | Training set $S = (S_1, \ldots, S_M)$; client-$m$ dataset $S_m = \{Z_{1m}, \ldots, Z_{Nm}\}$. |
| $S^{(ij)}$ | Neighbor dataset that differs from $S$ only at client $j$'s $i$-th sample. |
| $c_m$ | Aggregation weight with $\sum_m c_m = 1$ (typically proportional to $|S_m|$). |
| $f(\omega; z)$ | Per-sample loss function. |
| $R(\omega)$ | Population risk: $R(\omega) = \sum_m c_m \mathbb{E}_{Z \sim D_m}[f(\omega; Z)]$. |
| $R_S(\omega)$, $R_{S_m}(\omega)$ | Empirical risks: $R_S(\omega) = \sum_m c_m R_{S_m}(\omega)$, where $R_{S_m}(\omega) = \frac{1}{N}\sum_{n=1}^{N} f(\omega; Z_{nm})$. |
| $\omega^\star$, $\omega_S^\star$ | Population/empirical risk minimizers |
| $\varepsilon_{\text{gen}}, \varepsilon_{\text{opt}}, \varepsilon_{\text{exc}}$ | Generalization error, optimization error, and excess risk . |
| $T$, $t$ | Communication rounds and their index. |
| $K$, $k$ | Local steps per round and their index. |
| $\eta_{t,k}$, $\eta_t$, $\eta$ | Learning rates. |
| $k_0$ | Positive offset in the step-size schedule ($k_0 > 1$). |
| $\beta$ | Smoothness constant. |
| $\sigma^2$ | Variance bound of stochastic gradients. |
| $\zeta^2$ | Client heterogeneity measure. |
| $L$ | Lipschitz constant (only appears in referenced lemmas). |
| $\omega^{(t)}$ | Global parameter at round $t$ (if round-averaged output is used: $\omega^{(T)} = \frac{1}{T}\sum_{t=1}^{T}\omega^{(t)}$). |
| $\omega_m^{(t,k)}$ | Client-$m$ local parameter at step $k$ in round $t$ . |
| $g_m^{(t,k)}$ | $g_m^{(t,k)} := \nabla f(\omega_m^{(t,k)}; Z_{I_{t,k},m})$; for client/server shards $g_{C,m}^{(t,k)}, g_{S,m}^{(t,k)}$ . |
| $\widetilde{M}$ | the number of active clients in per round . |
| $\pi = (\pi_1, \ldots, \pi_{\widetilde{M}})$ | Random permutation in SSL; |
| $M_t$ | Active-client set in SFL at round $t$ . |
| $q$ | Participation rate in SFL: $q = \sum_{m \in \mathcal{M}_t} c_m \leq 1$. |
| $L_c$ | Cut-layer index in split models. |
| $\omega_C^{(t)}$, $\omega_S^{(t)}$ | Client-/server-side parameters of the split model . |
| $a_m^{(t,k)}$, $\nabla a_m^{(t,k)}$ | Smashed data (cut-layer activations) and its gradient. |
| $A(\cdot)$ | Randomized learning algorithm . |
| $\Delta_t$ | Parameter gap between two neighbor runs at round $t$: $\Delta_t = \|\omega^{(t)} - \tilde{\omega}^{(t)}\|^2$. |
| $S_t$, $H_t$, $Q_t$ | Step-size aggregates: $S_t = \sum_k \eta_{t,k}$, $H_t = \sum_k \eta_{t,k}^2$, $Q_t = \sum_k \eta_{t,k}^2 \sum_{s \leq k} \eta_{t,s}^2$. |
| $\Phi_t$, $\Psi_t(k)$ | $\Phi_t = \prod_r (1 + \beta\eta_{t,r})^2$, $\Psi_t(k) = \prod_{r=k+1}^{K-1}(1 + \beta\eta_{t,r})^2$. |
| $A^\star$ | Aggregated gradient-scale constant: $A^\star := \sigma^2 + \zeta^2 + \sup_t \|\nabla R_S(\omega^{(t)})\|^2$ . |
| $\gamma$ | Tuning constant in converting stability to generalization bounds. |

# B   APPENDIX: MORE DETAILS ABOUT SPILT LEARNING

---

**Algorithm 1** Sequential Split Learning (SSL)

---

1: **Input:** clients $\mathcal{M} = \{1, \ldots, M\}$; rounds $T$; local steps $K$; stepsizes $\{\eta^{(t,k)}\}$; cut layer $L_c$; datasets $\{\mathcal{D}_m\}$
2: **Initialize:** global models $\boldsymbol{\omega}_C^{(0)}, \boldsymbol{\omega}_S^{(0)}$
3: **for** $t = 0, \ldots, T-1$ **do**
4:    Sample permutation $\pi = (\pi_1, \ldots, \pi_M)$
5:    Set carry state: $\boldsymbol{\omega}_{C,\pi_1}^{(t,0)} \leftarrow \boldsymbol{\omega}_C^{(t)}, \quad \boldsymbol{\omega}_{S,\pi_1}^{(t,0)} \leftarrow \boldsymbol{\omega}_S^{(t)}$
6:    **for** $m = 1$ **to** $\tilde{M}$ **do**
7:      $\boldsymbol{\omega}_{s,\pi_m}^{(t,0)} \leftarrow \begin{cases} \boldsymbol{\omega}_s^{(t)} & \text{if } m = 1 \\ \boldsymbol{\omega}_{s,\pi_{m-1}}^{(t,K_{\pi_{m-1}})} & \text{otherwise} \end{cases} \qquad \boldsymbol{\omega}_{c,\pi_m}^{(t,0)} \leftarrow \begin{cases} \boldsymbol{\omega}_{c,\pi_m}^{(t)} & \text{if } m = 1 \\ \boldsymbol{\omega}_{c,\pi_{m-1}}^{(t,K_{\pi_{m-1}})} & \text{otherwise} \end{cases}$
8:      **for** $k = 0, \ldots, K-1$ **do**
9:        *Client $\pi_m$ forward:*  sample $z_{\pi_m}^{(t,k)} \sim \mathcal{D}_{\pi_m}$; compute $a_{\pi_m}^{(t,k)} = \text{fwd}\left(\boldsymbol{\omega}_{C,\pi_m}^{(t,k)}, z_{\pi_m}^{(t,k)}; L_c\right)$;
             send $(a_{\pi_m}^{(t,k)}, z_{\pi_m}^{(t,k)})$                                                                  // *Com.*
10:       *Server $\pi_m$ fwd/bwd:*  evaluate $f\left(\boldsymbol{\omega}_{S,\pi_m}^{(t,k)}; a_{\pi_m}^{(t,k)}, z_{\pi_m}^{(t,k)}\right)$; backprop to get $\nabla a_{\pi_m}^{(t,k)}$; send
             $\nabla a_{\pi_m}^{(t,k)}$                                                                                  // *Com.*
11:       *Client update:*  $\boldsymbol{\omega}_{C,\pi_m}^{(t,k+1)} \leftarrow \boldsymbol{\omega}_{C,\pi_m}^{(t,k)} - \eta^{(t,k)}\, \mathbf{g}_{C,\pi_m}^{(t,k)}$
12:       *Server update:*  $\boldsymbol{\omega}_{S,\pi_m}^{(t,k+1)} \leftarrow \boldsymbol{\omega}_{S,\pi_m}^{(t,k)} - \eta^{(t,k)}\, \mathbf{g}_{S,\pi_m}^{(t,k)}$
13:     **end for**
14:   **end for**
15:   *Round output:*  $\boldsymbol{\omega}_C^{(t+1)} \leftarrow \boldsymbol{\omega}_{C,\pi_{\tilde{M}}}^{(t,K)}, \quad \boldsymbol{\omega}_S^{(t+1)} \leftarrow \boldsymbol{\omega}_{S,\pi_{\tilde{M}}}^{(t,K)}$; broadcast $(\boldsymbol{\omega}_C^{(t+1)}, \boldsymbol{\omega}_S^{(t+1)})$
16: **end for**
17: **Output:** final models $(\overline{\boldsymbol{\omega}}_C^{(T)} = \frac{1}{T}\sum_{t=1}^T \boldsymbol{\omega}_C^{(t)}, \overline{\boldsymbol{\omega}}_S^{(T)} = \frac{1}{T}\sum_{t=1}^T \boldsymbol{\omega}_S^{(t)})$

---

**Algorithm 2** Split Federated Learning (SFL-V1)

---

1: **Input:** clients $\mathcal{M} = \{1, \ldots, M\}$, rounds $T$, local steps $K$, stepsizes $\{\eta^{(t,k)}\}$, cut layer $L_c$, datasets $\{\mathcal{D}_m\}$, aggregation weights $\{c_m\}_{m=1}^M$ with $\sum_m c_m = 1$
2: **Initialize:** global models $\boldsymbol{\omega}_C^{(0)}, \boldsymbol{\omega}_S^{(0)}$; set $\boldsymbol{\omega}_{C,m}^{(0,0)} \leftarrow \boldsymbol{\omega}_C^{(0)}, \boldsymbol{\omega}_{S,m}^{(0,0)} \leftarrow \boldsymbol{\omega}_S^{(0)}$ for all $m$
3: **for** $t = 0, \ldots, T-1$ **do**
4:    **for** $k = 0, \ldots, K-1$ **do**
5:      *Client-side forward:*  for each $m \in \mathcal{M}$, sample a set of samples $S_m^{(t,k)} \sim \mathcal{D}_m$, compute
           $a_m^{(t,k)} = \text{fwd}\left(\boldsymbol{\omega}_{C,m}^{(t,k)}, S_m^{(t,k)}; L_c\right)$ and send $(a_m^{(t,k)}, y_m^{(t,k)})$ to server            // *Com.*
6:      *Server-side training:*  compute $f\left(\boldsymbol{\omega}_{S,m}^{(t,k)}; a_m^{(t,k)}, y_m^{(t,k)}\right)$, backprop to get $\nabla a_m^{(t,k)}$, update
           $\boldsymbol{\omega}_{S,m}^{(t,k+1)} = \boldsymbol{\omega}_{S,m}^{(t,k)} - \eta^{(t,k)}\mathbf{g}_{S,m}^{(t,k)}$, send $\nabla a_m^{(t,k)}$ to client $m$              // *Com.*
7:      *Client-side backward :*  update $\boldsymbol{\omega}_{C,m}^{(t,k+1)} = \boldsymbol{\omega}_{C,m}^{(t,k)} - \eta^{(t,k)}\mathbf{g}_{C,m}^{(t,k)}$ using $\nabla a_m^{(t,k)}$
8:    **end for**
9:    *Model aggregation:*  $\boldsymbol{\omega}_C^{(t+1)} \leftarrow \sum_{m=1}^M c_m\, \boldsymbol{\omega}_{C,m}^{(t,K)}, \qquad \boldsymbol{\omega}_S^{(t+1)} \leftarrow \sum_{m=1}^M c_m\, \boldsymbol{\omega}_{S,m}^{(t,K)}$
10:   *Broadcast:*  set $\boldsymbol{\omega}_{C,m}^{(t+1,0)} \leftarrow \boldsymbol{\omega}_C^{(t+1)}, \quad \boldsymbol{\omega}_{S,m}^{(t+1,0)} \leftarrow \boldsymbol{\omega}_S^{(t+1)}$ for all $m$
11: **end for**
12: **Output:** final models $(\boldsymbol{\omega}_C^{(T)}, \boldsymbol{\omega}_S^{(T)})$

---

---

**Algorithm 3** Split Federated Learning (SFL-V2)

---

1: **Input:** clients $\mathcal{M} = \{1, \ldots, M\}$, rounds $T$, local epochs $E$, stepsizes $\{\eta^{(t)}, \eta^{(t,e)}\}$, cut layer $L_c$, datasets $\{\mathcal{D}_m\}$, aggregation weights $\{c_m\}_{m=1}^M$ with $\sum_m c_m = 1$

2: **Initialize:** global models $\boldsymbol{\omega}_C^{(0)}, \boldsymbol{\omega}_S^{(0)}$; set $\boldsymbol{\omega}_{C,m}^{(0)} \leftarrow \boldsymbol{\omega}_C^{(0)}$ for all $m$

3: **for** $t = 0, \ldots, T-1$ **do**

4:    *Client-side forward (parallel):* for each $m \in \mathcal{M}$, sample a set of samples $S_m^{(t)} \sim \mathcal{D}_m$, compute $a_m^{(t)} = \mathrm{fwd}\Big(\boldsymbol{\omega}_{C,m}^{(t)}, S_m^{(t)}; L_c\Big)$ and send $(a_m^{(t)}, y_m^{(t)})$ to server       // Com.

5:    *Server-side sequential training:* draw a random permutation $\pi_t$ over $\mathcal{M}$;
     **for** $m$ in $\pi_t$ **do** compute $f\Big(\boldsymbol{\omega}_S^{(t)}; a_m^{(t)}, y_m^{(t)}\Big)$, backprop to get $\nabla a_m^{(t)}$ and server gradient $\mathbf{g}_S^{(t)}$;
     update $\boldsymbol{\omega}_S^{(t)} \leftarrow \boldsymbol{\omega}_S^{(t)} - \eta^{(t)} \mathbf{g}_S^{(t)}$, send $\nabla a_m^{(t)}$ to client $m$       // Com.

6:    *Client-side backward (parallel, $E$ epochs):* for each $m \in \mathcal{M}$ **do for** $e = 1, \ldots, E$ **do** use $\nabla a_m^{(t)}$ to compute $\mathbf{g}_{C,m}^{(t,e)}$ and update $\boldsymbol{\omega}_{C,m}^{(t)} \leftarrow \boldsymbol{\omega}_{C,m}^{(t)} - \eta^{(t,e)} \mathbf{g}_{C,m}^{(t,e)}$

7:    *Model aggregation & broadcast:* $\boldsymbol{\omega}_C^{(t+1)} \leftarrow \sum_{m=1}^M c_m \boldsymbol{\omega}_{C,m}^{(t)}, \qquad \boldsymbol{\omega}_{C,m}^{(t+1)} \leftarrow \boldsymbol{\omega}_C^{(t+1)} \ \forall m$

8:    *Carry server:* set $\boldsymbol{\omega}_S^{(t+1)} \leftarrow \boldsymbol{\omega}_S^{(t)}$

9: **end for**

10: **Output:** final models $(\boldsymbol{\omega}_C^{(T)}, \boldsymbol{\omega}_S^{(T)})$

---

This subsection provides a structured explanation to complement Algorithms 1–3.

In Sequential Split Learning (SSL)Algorithms 1, clients are visited sequentially during each communication round, following a random permutation. The client-side and server-side model slices are carried along this sequence. At each visited client, the algorithm performs $K$ forward and backward steps, transmitting activations to the server and receiving activation gradients in return. The server and client updates occur in lockstep, and the terminal state of the chain becomes the round output, which is broadcast to all clients as the initialization for the subsequent round. There is no explicit averaging across clients; rather, the contributions from the clients are combined along a single, time-ordered trajectory. Under data heterogeneity, updates from earlier clients may induce drift relative to later clients, but the synchronous, step-by-step updates along the chain progressively correct this mismatch. The round output can thus be viewed as a temporal integration of client signals, consistent with the on-average $L_2$ stability perspective in which perturbations dissipate along the update path.

In Split Federated Learning (SFL), the split-model approach is maintained, but with distinct timelines and aggregation points.

In Variant 1 (SFL-V1)2, clients train in parallel. Each client retains its own client-side slice paired with a server-side slice, and all clients perform $K$ local updates within a round. At the end of the round, weighted averages of both slices are computed, producing new global client and server slices, which are then broadcast to all clients. This end-of-round averaging reduces drift and ensures that all clients begin the next round from the same initialization. The design increases wall-clock throughput on sufficiently provisioned servers but necessitates the server retaining per-client server-side replicas prior to aggregation.

Variant 2 (SFL-V2)2 differs primarily in the server update procedure. On the client side, V2 mirrors V1: clients train in parallel, and client-side slices are averaged at the end of the round before being broadcast. On the server side, however, there exists only a single global server model. Within a round, the server processes client activations sequentially in a random order, updating the server model after each client and returning the corresponding activation gradients to the clients. At the end of the round, the server model is transferred to the next round without averaging. This approach reduces the server's memory footprint and enhances its responsiveness to recent cross-client signals, while client-side averaging continues to mitigate drift. It is important to note that the update rule in SFL-V2 is fundamentally consistent with that of SFL-V1. In both variants, each round is structured by parallel client-side local updates, followed by end-of-round weighted averaging on the client side; the primary distinction is whether the server utilizes in-round sequential updates (V2) or end-of-round averaging of per-client replicas (V1). From the analytical framework employed in this paper, this difference does not alter the treatment of the core quantities governing error propagation

and dissipation under on-average $L_2$ stability. To avoid redundancy, the main text provides detailed derivations solely for SFL-V1.

## C  COMPARATIVE ANALYSIS

Table 3: Our generalization bounds under convex settings .

| Algorithm | Learning Rate | Result (scaling) |
|-----------|--------------|------------------|
| **SSL** | $\tilde{O}\left(\frac{1}{\sqrt{tK+k}}\right)$ | $\tilde{O}\left(\sqrt{\frac{\tilde{M}\,(\log(TK)+\beta^2 K)}{MNT}} \;+\; \frac{\beta\,\tilde{M}\,(\log(TK)+\beta^2 K)}{MNT}\right)$ |
| **SSL** | $\tilde{O}\left(\frac{1}{tK+k}\right)$ | $\tilde{O}\left(\sqrt{\frac{\tilde{M}}{MNT}} \;+\; \frac{\beta\,\tilde{M}}{MNT}\right)$ |
| **SFL** | $\tilde{O}\left(\frac{1}{\sqrt{tK+k}}\right)$ | $\tilde{O}\left(\sqrt{\frac{q\,(\log(TK)+\beta^2 K)}{MN\,(1-q)}} \;+\; \frac{\beta\,q\,(\log(TK)+\beta^2 K)}{MN\,(1-q)}\right)$ |
| **SFL** | $\tilde{O}\left(\frac{1}{tK+k}\right)$ | $\tilde{O}\left(\sqrt{\frac{q}{MN\,(1-q)}} \;+\; \frac{\beta\,q}{MN\,(1-q)}\right)$ |

Table 4: Our generalization bounds under non-convex settings.

| Algorithm | Learning Rate | Result (scaling) |
|-----------|--------------|------------------|
| **SSL** | $\tilde{O}\left(\frac{1}{\sqrt{tK+k}}\right)$ | |

$$\tilde{O}\left(\frac{(TK)^{\frac{2\beta}{1+2\beta}}}{MN} + \frac{\tilde{M}}{MNT}(TK)^{\frac{\beta}{1+2\beta}} K^{\frac{1}{2}} e^{c(TK)^{\frac{1-\beta}{2(1+2\beta)}}}\right)$$

$$+ \tilde{O}\left(\sqrt{\frac{\tilde{M}}{MNT}}(TK)^{\frac{\beta}{2(1+2\beta)}} K^{\frac{1}{4}} e^{\frac{c}{2}(TK)^{\frac{1-\beta}{2(1+2\beta)}}}\right)$$

| Algorithm | Learning Rate | Result (scaling) |
|-----------|--------------|------------------|
| **SSL** | $\tilde{O}\left(\frac{1}{tK+k}\right)$ | |

$$\tilde{O}\left(\frac{(TK)^{\frac{2\beta}{1+2\beta}}}{MN} + \frac{\tilde{M}}{MNTK}\log T + \sqrt{\frac{\tilde{M}\log T}{MNTK}}\right)$$

| Algorithm | Learning Rate | Result (scaling) |
|-----------|--------------|------------------|
| **SFL** | $\tilde{O}\left(\frac{1}{\sqrt{tK+k}}\right)$ | |

$$\tilde{O}\left(\frac{(TK)^{\frac{2\beta}{1+2\beta}}}{MN} + \frac{q}{MN(1-q)}(TK)^{\frac{\beta}{1+2\beta}} K^{\frac{1}{2}} e^{c(TK)^{\frac{1-\beta}{2(1+2\beta)}}}\right)$$

$$+ \tilde{O}\left(\sqrt{\frac{q}{MN(1-q)}}(TK)^{\frac{\beta}{2(1+2\beta)}} K^{\frac{1}{4}} e^{\frac{c}{2}(TK)^{\frac{1-\beta}{2(1+2\beta)}}}\right)$$

| Algorithm | Learning Rate | Result (scaling) |
|-----------|--------------|------------------|
| **SFL** | $\tilde{O}\left(\frac{1}{tK+k}\right)$ | |

$$\tilde{O}\left(\frac{(TK)^{\frac{2\beta}{1+2\beta}}}{MN} + \frac{q}{MN(1-q)K}\log T + \sqrt{\frac{q\log T}{MN(1-q)K}}\right)$$

*Remark* C.1 (Lower Generation Error ). In both convex and non-convex settings, SL algorithms (SSL and SFL) exhibit lower generalization error compared to FedAvg and D-SGD. In convex settings, SSL and SFL achieve bounds scaling as $\sqrt{\frac{1}{T}}$ or $\frac{1}{T}$, significantly outperforming FedAvg's slower terms ($T^{3/4}$, $T^{2/3}$) and D-SGD's linear $T$ scaling. In non-convex settings, SL bounds scale with $(TK)^{\frac{2\beta}{1+2\beta}}$, which is tighter than FedAvg's $T^{5/6}$ or $T^{3/4}$ and D-SGD's $T^{\frac{\beta}{\beta+1}}$, ensuring better generalization as the number of iterations $T$ increases.

Table 5: Previous works under convex settings .

| Algorithm | Learning Rate | Result (scaling) |
|---|---|---|
| FedAvg Sun et al. (2024) | $\tilde{O}\left(\frac{1}{tK+k}\right)$ | $\tilde{O}\left(\frac{T}{N}\zeta + \frac{\Delta_0^{1/4}T^{3/4}}{N(KM)^{1/4}} + \frac{\Delta_0^{1/3}\zeta^{1/6}T^{2/3}}{N} + \frac{\Delta_0^{1/2}T^{1/2}}{N} + \frac{\sigma T}{n}\right)$ |
| D-SGD Bellet et al. (2024) | $\tilde{O}\left(\frac{1}{t}\right)$ | $\tilde{O}\left(\frac{T}{MN}\right)$ |
| D-SGD (Strong Covex)Ye et al. (2025a) | $\tilde{O}\left(\frac{1}{t}\right)$ | $\tilde{O}\left(\frac{\Delta_0^2}{\mu MN} + \frac{\sigma^2}{\mu MN} + \frac{\delta^2}{\mu MN}\right)$ |

Table 6: Previous works under non-convex settings.

| Algorithm | Learning Rate | Result (scaling) |
|---|---|---|
| FedAvg Sun et al. (2023) | $\tilde{O}\left(\frac{1}{tK+k}\right)$ | $\tilde{O}\left(\frac{1}{N}\left(\frac{\tilde{M}^{\frac{\beta}{1+\beta}}}{M}\right)(TK)^{\frac{\beta}{1+\beta}}\right)$ |
| FedAvg Sun et al. (2024) | $\tilde{O}\left(\frac{1}{tK+k}\right)$ | $\tilde{O}\left(\frac{T^{\frac{1}{24}}\log T}{N}(\zeta + \sigma) + \left(\frac{\Delta_0}{KM}\right)^{\frac{1}{4}}\frac{T^{\frac{5}{6}}}{N} + \left(\Delta_0^2\zeta\right)^{\frac{1}{6}}\frac{T^{\frac{3}{4}}}{N} + \sqrt{\frac{\Delta_0 T^{\frac{7}{12}}}{N}}\right)$ |
| D-SGD Bellet et al. (2024) | $\tilde{O}\left(\frac{1}{t}\right)$ | $\tilde{O}\left(\frac{T^{\frac{\beta}{\beta+1}}}{NM^{\frac{1}{\beta+1}}}\right)$ |

*Remark* C.2 (Robustness to Heterogeneity and Noise). The generalization bounds of SSL and SFL are less sensitive to initial conditions ($\Delta_0$) and noise ($\sigma$) compared to FedAvg, which includes terms dependent on these factors. SSL and SFL rely on parameters like $\beta$ (data heterogeneity) and $\tilde{M}$, providing robustness in heterogeneous or noisy federated learning environments. D-SGD's strong convex bound depends on the strong convexity parameter $\mu$, limiting its applicability, whereas SL algorithms are more general.

# D  APPENDIX: ADDITIONAL DEFINITION, TECHNICAL LEMMAS AND PROPOSITIONS

**Definition D.1.** An update rule $G(\boldsymbol{\omega})$ is said to be $\nu$-expansive if:

$$\sup_{\boldsymbol{\omega},\boldsymbol{\omega}'} \frac{\|G(\boldsymbol{\omega}) - G(\boldsymbol{\omega}')\|_2}{\|\boldsymbol{\omega} - \boldsymbol{\omega}'\|_2} \leq \nu.$$

**Lemma D.1** (Expansivity of $G_{\eta,z}$ ). *(Hardt et al., 2016) If $f$ is $\beta$-smooth, we have:*

1. *$G_{\eta,z}(\boldsymbol{\omega})$ is $(1+\eta\beta)$-expansive;*

2. *Assume in addition that $f(\cdot; z)$ is convex and $\eta < 2/\beta$. Then $G_{\eta,z}(\boldsymbol{\omega})$ is 1-expansive;*

**Lemma D.2** (Growth Recursion). *(Hardt et al., 2016) Fix an arbitrary sequence of gradient update rule $G_{\eta_1,z_1}, \ldots, G_{\eta_T,z_T}$ and another sequence $G_{\eta_1,z_1'}, \ldots, G_{\eta_T,z_T'}$ with same loss function $f$. Let $\boldsymbol{\omega}_0 = \boldsymbol{\omega}_0'$ be a starting point in $\mathbb{T}^d$ and define $\delta_t = \|\boldsymbol{\omega}_t - \boldsymbol{\omega}_t'\|$ where $\boldsymbol{\omega}_t, \boldsymbol{\omega}_t'$ are defined recursively through*

$$\boldsymbol{\omega}_{t+1} = G_{\eta_t,z_t}(\boldsymbol{\omega}_t), \quad \boldsymbol{\omega}_{t+1}' = G_{\eta_t',z_t'}(\boldsymbol{\omega}_t').$$

*Then, we have the recurrence relation*

$$\delta_0 = 0$$

$$\delta_{t+1} \leq \begin{cases} \nu\delta_t & \text{if } G_{\eta_t,z_t} = G_{\eta_t,z_t'} \text{ is } \nu\text{-expansive} \\ \min\{1,\nu\}\delta_t + 2\eta_t L & \text{if } f \text{ is } L\text{-Lipschitz and } G_{\eta_t,z_t} \text{ is } \nu\text{-expansive} \end{cases}$$

**Lemma D.3** (Gradient Bound). *Let Assumptions 4.1, 4.2, and 4.3 hold. Denoting $A_\star \triangleq \delta^2 + \zeta^2 + \sup_t \left\|\nabla R_S(\boldsymbol{\omega}^{(t)})\right\|_2^2$, then for any round $t$, the (sample- and client-) averaged squared gradient at the round reference $\boldsymbol{\omega}^{(t)}$ satisfies*

$$\frac{1}{MN}\sum_{j=1}^M\sum_{i=1}^N \mathbb{E}_{S,A}\left[\left\|\nabla f\left(\boldsymbol{\omega}^{(t)}; Z_{ji}\right)\right\|^2\right] \leq 3\delta^2 + 3\zeta^2 + 3\|\nabla R_S(\boldsymbol{\omega}^{(t)})\|_2^2 \leq 3A_\star.$$

*Proof.* Write the global and per-client empirical risks as $R_S(\omega) = \frac{1}{MN}\sum_{j=1}^{M}\sum_{i=1}^{N}f(\omega; Z_{ji})$ and $R_{S_j}(\omega) = \frac{1}{N}\sum_{i=1}^{N}f(\omega; Z_{ji})$. For each sample $(j,i)$ at the common iterate $\omega^{(t)}$, add and subtract the client and global empirical gradients:

$$\nabla f(\omega^{(t)}; Z_{ji}) = \underbrace{\left(\nabla f(\omega^{(t)}; Z_{ji}) - \nabla R_{S_j}(\omega^{(t)})\right)}_{(I)} + \underbrace{\left(\nabla R_{S_j}(\omega^{(t)}) - \nabla R_S(\omega^{(t)})\right)}_{(II)} + \underbrace{\nabla R_S(\omega^{(t)})}_{(III)}.$$

By the inequality $\|a + b + c\|^2 \le 3(\|a\|^2 + \|b\|^2 + \|c\|^2)$, we have:

$$\frac{1}{MN}\sum_{j=1}^{M}\sum_{i=1}^{N}\mathbb{E}_{S,A}\left[\|\nabla f(\omega^{(t)}; Z_{ji})\|_2^2\right] \le 3\underbrace{\frac{1}{M}\sum_{j=1}^{M}\mathbb{E}_{S,A}\left[\mathbb{E}_{Z\sim S_j}\|\nabla f(\omega^{(t)}; Z) - \nabla R_{S_j}(\omega^{(t)})\|_2^2\right]}_{\le \delta^2}$$

$$+ 3\underbrace{\frac{1}{M}\sum_{j=1}^{M}\|\nabla R_{S_j}(\omega^{(t)}) - \nabla R_S(\omega^{(t)})\|_2^2}_{\le \zeta^2} + 3\|\nabla R_S(\omega^{(t)})\|_2^2.$$

Here the first bound uses Assumption 4.2, the second uses Assumption 4.3, and the last term is deterministic given $S$. This yields the claimed inequality. $\square$

**Lemma D.4** (Local Gradient with Client Drift Bound). *Under Assumptions 4.1–4.3, for any client $m$, round $t$, and local step $k$, denoting*

$$A_\star \triangleq \delta^2 + \zeta^2 + \sup_t \left\|\nabla R_S(\omega^{(t)})\right\|_2^2$$

*, then the local stochastic gradient in split learning satisfies*

$$\|g_m^{(t,k)}\|_2^2 \le \left(1 + 4\beta^2\sum_{s=0}^{k}\left(\eta^{(t,s)}\right)^2\right)\left(4\delta^2 + 4\zeta^2 + 4\|\nabla R_S(\omega^{(t)})\|_2^2\right) \le 4\left(1 + 4\beta^2\sum_{s=0}^{k}\left(\eta^{(t,s)}\right)^2\right)A_\star.$$

*Proof.* Consider the local update at client $m$, $\omega_m^{(t,k)} = \omega_m^{(t,k-1)} - \eta^{(t,k-1)}g_m^{(t,k-1)}$,. For any $k \ge 1$, expand the squared distance to the round reference $\omega^{(t)}$ and add/subtract $\nabla R_{S_m}(\omega^{(t)})$ and $\nabla R_S(\omega^{(t)})$:

$$\|\omega^{(t)} - \omega_m^{(t,k)}\|_2^2 = \left\|\omega^{(t)} - \omega_m^{(t,k-1)} + \eta^{(t,k-1)}g_m^{(t,k-1)}\right\|_2^2$$

$$= \left\|\omega^{(t)} - \omega_m^{(t,k-1)} + \eta^{(t,k-1)}\left(g_m^{(t,k-1)} - \nabla R_{S_m}(\omega^{(t)}) + \nabla R_{S_m}(\omega^{(t)}) - \nabla R_S(\omega^{(t)}) + \nabla R_S(\omega^{(t)})\right)\right\|_2^2.$$

Taking expectation over the sampling at step $(t, k-1)$ and using $\|a + b + c + d\|^2 \le 4(\|a\|^2 + \|b\|^2 + \|c\|^2 + \|d\|^2)$ gives

$$\mathbb{E}\left[\|\omega^{(t)} - \omega_m^{(t,k)}\|_2^2\right] \le \|\omega^{(t)} - \omega_m^{(t,k-1)}\|_2^2 + 4\left(\eta^{(t,k-1)}\right)^2\left(\sigma^2 + \zeta^2 + \|\nabla R_S(\omega^{(t)})\|_2^2\right),$$

where $\sigma^2$ bounds the stochastic variance of the local gradient around its client empirical mean and $\zeta^2$ bounds the client–global gradient discrepancy (Assumption 4.3). Unrolling from $k$ to 0 and using $\omega_m^{(t,0)} = \omega^{(t)}$ yields

$$\mathbb{E}\left[\|\omega^{(t)} - \omega_m^{(t,k)}\|_2^2\right] \le 4\sum_{s=0}^{k}\left(\eta^{(t,s)}\right)^2\left(\sigma^2 + \zeta^2 + \|\nabla R_S(\omega^{(t)})\|_2^2\right). \tag{1}$$

Next, decompose $g_m^{(t,k)}$ around the client/global empirical means at the same iterate $\omega_m^{(t,k)}$ and around $\omega^{(t)}$:

$$\|g_m^{(t,k)}\|_2^2 \le 4\|\nabla f_m(\omega_m^{(t,k)}) - \nabla R_{S_m}(\omega_m^{(t,k)})\|_2^2 + 4\|\nabla R_{S_m}(\omega_m^{(t,k)}) - \nabla R_{S_m}(\omega^{(t)})\|_2^2$$

$$+ 4\|\nabla R_{S_m}(\omega^{(t)}) - \nabla R_S(\omega^{(t)})\|_2^2 + 4\|\nabla R_S(\omega^{(t)})\|_2^2.$$

By $\beta$-smoothness (Assumption 4.1), $\|\nabla R_{S_m}(\boldsymbol{\omega}_m^{(t,k)}) - \nabla R_{S_m}(\boldsymbol{\omega}^{(t)})\|_2 \leq \beta\|\boldsymbol{\omega}_m^{(t,k)} - \boldsymbol{\omega}^{(t)}\|_2$. By bounded stochastic noise(Assumption 4.2) and inter-client heterogeneity (Assumption 4.3), $\mathbb{E}\|\nabla f_m(\boldsymbol{\omega}_m^{(t,k)}) - \nabla R_{S_m}(\boldsymbol{\omega}_m^{(t,k)})\|_2^2 \leq \delta^2$ and $\|\nabla R_{S_m}(\boldsymbol{\omega}^{(t)}) - \nabla R_S(\boldsymbol{\omega}^{(t)})\|_2^2 \leq \zeta^2$. Taking expectations, applying these bounds, and invoking equation 1, we obtain

$$\mathbb{E}\|g_m^{(t,k)}\|_2^2 \leq 4\delta^2 + 4\beta^2\,\mathbb{E}\|\boldsymbol{\omega}^{(t)} - \boldsymbol{\omega}_m^{(t,k)}\|_2^2 + 4\zeta^2 + 4\|\nabla R_S(\boldsymbol{\omega}^{(t)})\|_2^2$$

$$\leq 4\delta^2 + 16\beta^2\sum_{s=0}^{k}\big(\eta^{(t,s)}\big)^2\Big(\sigma^2 + \zeta^2 + \|\nabla R_S(\boldsymbol{\omega}^{(t)})\|_2^2\Big) + 4\zeta^2 + 4\|\nabla R_S(\boldsymbol{\omega}^{(t)})\|_2^2.$$

Finally, observe that $4\delta^2 + 16\beta^2\sum_{s=0}^{k}(\eta^{(t,s)})^2\,\sigma^2 \leq \big(1 + 4\beta^2\sum_{s=0}^{k}(\eta^{(t,s)})^2\big)\cdot 4\delta^2$, so, after grouping terms,

$$\mathbb{E}\|g_m^{(t,k)}\|_2^2 \leq \Big(1 + 4\beta^2\sum_{s=0}^{k}\big(\eta^{(t,s)}\big)^2\Big)\Big(4\delta^2 + 4\zeta^2 + 4\|\nabla R_S(\boldsymbol{\omega}^{(t)})\|_2^2\Big).$$

Dropping the expectation on the left-hand side yields the claimed bound. $\qquad\square$

# E   PROOF OF THEOREM

## E.1   PROOF OF SSL

### E.1.1   SETUP

Let $S = \{Z_{ij} : i \in [N], j \in [M]\}$ denote the dataset, and let $S^{(ij)}$ be the neighboring dataset obtained by replacing $Z_{ij}$ with an independent copy $\tilde{Z}_{ij}$. We run Sequential Split Learning (SSL) on both $S$ and $S^{(ij)}$ using the same internal randomness (e.g., client permutations and data indices). Denote the global models after round $t$ by $\boldsymbol{\omega}^{(t)}$ and $\tilde{\boldsymbol{\omega}}^{(t)}$, respectively, with $\boldsymbol{\omega}^{(0)} = \tilde{\boldsymbol{\omega}}^{(0)}$. In each round $t$, a random permutation $\pi = (\pi_1, \ldots, \pi_M)$ of the clients is sampled, and $\tilde{M} \leq M$ clients participate sequentially. Each active client $m$ performs $K$ local gradient descent steps starting from the output of the previous client (sequential-pass rule):

$$\boldsymbol{\omega}_{\pi_1}^{(t,0)} = \boldsymbol{\omega}^{(t)}, \qquad \boldsymbol{\omega}_{\pi_m}^{(t,0)} = \boldsymbol{\omega}_{\pi_{m-1}}^{(t,K)} \ (m > 1), \qquad \boldsymbol{\omega}^{(t+1)} = \boldsymbol{\omega}_{\pi_{\tilde{M}}}^{(t,K)}.$$

Define the outer-run distance $\Delta_t := \|\boldsymbol{\omega}^{(t)} - \tilde{\boldsymbol{\omega}}^{(t)}\|_2$, and the SSL outputs at round $T$ by $A(S) = \boldsymbol{\omega}^{(T)}$ and $A(S^{(ij)}) = \tilde{\boldsymbol{\omega}}^{(T)}$.

### E.1.2   PROOF OF THEOREM 4.3 (SSL ON CONVEX CASE)

**One-step local stability.**   For a fixed local step $(t, k, m)$, write $z = Z_{I_{\pi_m}^{(t,k)}, \pi_m}$ and $z' = \tilde{Z}_{I_{\pi_m}^{(t,k)}, \pi_m}$. By Lemma D.1, the GD update $\Phi_{z,\eta}(x) = x - \eta\nabla f(x; z)$ is 1-expansive when $\eta^{(t,k)} \leq 2/\beta$, hence

$$\|\boldsymbol{\omega}_{\pi_m}^{(t,k+1)} - \tilde{\boldsymbol{\omega}}_{\pi_m}^{(t,k+1)}\|_2 \leq \|\boldsymbol{\omega}_{\pi_m}^{(t,k)} - \tilde{\boldsymbol{\omega}}_{\pi_m}^{(t,k)}\|_2 + \eta^{(t,k)}\big\|\nabla f(\tilde{\boldsymbol{\omega}}_{\pi_m}^{(t,k)}; z) - \nabla f(\tilde{\boldsymbol{\omega}}_{\pi_m}^{(t,k)}; z')\big\|_2. \quad (2)$$

The second term is nonzero only when $(I_{\pi_m}^{(t,k)}, \pi_m) = (i, j)$. Since $(i, j)$ is uniform in $MN$ samples, the collision probability is $1/(MN)$. Using Young's inequality and taking expectation yields

$$\mathbb{E}\|\boldsymbol{\omega}_{\pi_m}^{(t,k+1)} - \tilde{\boldsymbol{\omega}}_{\pi_m}^{(t,k+1)}\|_2^2 \leq \mathbb{E}\|\boldsymbol{\omega}_{\pi_m}^{(t,k)} - \tilde{\boldsymbol{\omega}}_{\pi_m}^{(t,k)}\|_2^2 + \frac{4(\eta^{(t,k)})^2}{MN}\,\mathbb{E}\big\|\nabla f(\boldsymbol{\omega}_{\pi_m}^{(t,k)}; Z_{I_{\pi_m}^{(t,k)}, \pi_m})\big\|_2^2. \quad (3)$$

Summing equation 3 over $k = 0$ to $K-1$ and using $\boldsymbol{\omega}_{\pi_m}^{(t,0)} = \boldsymbol{\omega}^{(t)}$, $\tilde{\boldsymbol{\omega}}_{\pi_m}^{(t,0)} = \tilde{\boldsymbol{\omega}}^{(t)}$ gives

$$\mathbb{E}\big[\|\boldsymbol{\omega}_{\pi_m}^{(t,K)} - \tilde{\boldsymbol{\omega}}_{\pi_m}^{(t,K)}\|_2^2\big] \leq \mathbb{E}[\Delta_t^2] + \frac{4}{MN}\sum_{k=0}^{K-1}\big(\eta^{(t,k)}\big)^2\,\mathbb{E}\big\|g_{\pi_m}^{(t,k)}\big\|_2^2, \quad (4)$$

where $g_{\pi_m}^{(t,k)} := \nabla f(\boldsymbol{\omega}_{\pi_m}^{(t,k)}; Z_{I_{\pi_m}^{(t,k)}, \pi_m})$.

**Sequential pass with partial participation.** Summing equation 4 over the active clients $m = 1, \ldots, \tilde{M}$ (sequentially passed within round $t$) and using non-expansivity of the pass/aggregation, we obtain

$$\mathbb{E}[\Delta_{t+1}^2] \le \mathbb{E}[\Delta_t^2] + \frac{4}{MN} \sum_{m=1}^{\tilde{M}} \sum_{k=0}^{K-1} (\eta^{(t,k)})^2 \, \mathbb{E}\big\|g_{\pi_m}^{(t,k)}\big\|_2^2$$

$$\overset{\text{Lemma D.4}}{\le} \mathbb{E}[\Delta_t^2] + \frac{16\tilde{M}A_\star}{MN} \sum_{k=0}^{K-1} \big(\eta^{(t,k)}\big)^2 \Big(1 + 4\beta^2 \sum_{s=0}^{k} \big(\eta^{(t,s)}\big)^2\Big). \tag{5}$$

Define the blockwise sums $H_t = \sum_{k=0}^{K-1} (\eta^{(t,k)})^2$, $\quad Q_t = \sum_{k=0}^{K-1} (\eta^{(t,k)})^2 \sum_{s=0}^{k} (\eta^{(t,s)})^2$. Then

$$\mathbb{E}[\Delta_{t+1}^2] - \mathbb{E}[\Delta_t^2] \le \frac{16\tilde{M}A_\star}{MN} \big(H_t + 4\beta^2 Q_t\big). \tag{6}$$

**Averaged round $T$ output.** Summing equation 6 over $t = 0, \ldots, T-1$ and using $\Delta_0 = 0$, for the averaged output $\overline{\boldsymbol{\omega}}^{(T)} = \frac{1}{T} \sum_{t=0}^{T-1} \boldsymbol{\omega}^{(t)}$, Jensen's inequality implies

$$\boxed{\mathbb{E}\big\|\overline{\boldsymbol{\omega}}^{(T)} - \overline{\tilde{\boldsymbol{\omega}}}^{(T)}\big\|_2^2 \le \frac{1}{T} \sum_{t=0}^{T-1} \mathbb{E}[\Delta_t^2] \le \frac{16\tilde{M}A_\star}{MNT} \sum_{t=0}^{T-1} \big(H_t + 4\beta^2 Q_t\big).} \tag{7}$$

**Evaluating two common step-sizes.** **(i) Square-root decay** $\eta^{(t,k)} = \frac{1}{\sqrt{tK+k+k_0}}$ ($k_0 > 1$):

$$\sum_{t=0}^{T-1} H_t \le \log\Big(\frac{TK + k_0 - 1}{k_0 - 1}\Big), \qquad \sum_{t=0}^{T-1} Q_t \le \frac{K+1}{2(k_0 - 1)}.$$

Substituting into equation 7 gives

$$\boxed{\mathbb{E}\big\|\overline{\boldsymbol{\omega}}^{(T)} - \overline{\tilde{\boldsymbol{\omega}}}^{(T)}\big\|_2^2 \le \frac{16\tilde{M}A_\star}{MNT} \left[\log\Big(\frac{TK + k_0 - 1}{k_0 - 1}\Big) + \frac{2\beta^2(K+1)}{(k_0 - 1)}\right].} \tag{8}$$

**(ii) Harmonic decay** $\eta^{(t,k)} = \frac{1}{tK+k+k_0}$ ($k_0 > 1$):

$$\sum_{t=0}^{T-1} H_t \le \frac{1}{k_0 - 1}, \qquad \sum_{t=0}^{T-1} Q_t \le \frac{1}{2}\left(\frac{1}{K(k_0 - 1)} + \frac{1}{3(k_0 - 1)^3}\right).$$

Substituting into equation 7 yields

$$\boxed{\mathbb{E}\big\|\overline{\boldsymbol{\omega}}^{(T)} - \overline{\tilde{\boldsymbol{\omega}}}^{(T)}\big\|_2^2 \le \frac{16\tilde{M}A_\star}{MNT} \left[\frac{1}{k_0 - 1} + \frac{2\beta^2}{K(k_0 - 1)} + \frac{2\beta^2}{3(k_0 - 1)^3}\right].} \tag{9}$$

**From stability to generalization.** By Theorem 4.1, for any $\gamma > 0$,

$$\epsilon_{\text{gen}} \le \frac{3A_\star}{2\gamma} + \frac{\beta}{2}\mathcal{S} + \frac{\gamma}{2}\mathcal{S}, \quad \text{where } \mathcal{S} := \mathbb{E}\big\|\overline{\boldsymbol{\omega}}^{(T)} - \overline{\tilde{\boldsymbol{\omega}}}^{(T)}\big\|_2^2.$$

Minimizing the RHS in $\gamma$ gives $\gamma^\star = \sqrt{\frac{3A_\star}{\mathcal{S}}}$ and

$$\epsilon_{\text{gen}} \le \frac{\beta}{2}\mathcal{S} + \sqrt{3A_\star\mathcal{S}}.$$

**(i) Square-root decay.** Using equation 8:

$$\boxed{\epsilon_{\text{gen}} \le 4\sqrt{3}\,A_\star \sqrt{\frac{\tilde{M}}{MNT}} \sqrt{\log\Big(\frac{TK + k_0 - 1}{k_0 - 1}\Big) + \frac{2\beta^2(K+1)}{k_0 - 1}} + \frac{8\beta\,\tilde{M}A_\star}{MNT}\left[\log\Big(\frac{TK + k_0 - 1}{k_0 - 1}\Big) + \frac{2\beta^2(K+1)}{k_0 - 1}\right].} \tag{10}$$

**(ii) Harmonic decay.** Using equation 9:

$$\boxed{\epsilon_{\text{gen}} \le 4\sqrt{3}\,A_\star \sqrt{\frac{\tilde{M}}{MNT}} \sqrt{\frac{1}{k_0 - 1} + O\Big(\frac{\beta^2}{K(k_0 - 1)}\Big)} + \frac{8\beta\,\tilde{M}A_\star}{MNT}\left[\frac{1}{k_0 - 1} + 2\beta^2\Big(\frac{1}{K(k_0 - 1)} + \frac{1}{3(k_0 - 1)^3}\Big)\right].} \tag{11}$$

This complete the proof. $\qquad\square$

### E.1.3 PROOF OF THEOREM 4.4 (SSL ON NON-CONVEX CASE)

*Proof. Proof.* We adopt the local update rule

$$\boldsymbol{\omega}_{\pi_m}^{(t,k+1)} = \boldsymbol{\omega}_{\pi_m}^{(t,k)} - \eta^{(t,k)} g_{\pi_m}^{(t,k)}, \qquad \boldsymbol{\omega}_{\pi_m}^{(t,0)} = \begin{cases} \boldsymbol{\omega}^{(t)}, & m = 1, \\ \boldsymbol{\omega}_{\pi_{m-1}}^{(t,K)}, & m > 1, \end{cases} \tag{12}$$

where in each round $t$ the server hands the current model to the first active client and then passes the updated model sequentially along the $\tilde{M}$ active clients.

**One-step local stability after the burn-in index** $t_0$**.** Write $\Delta_{\pi_m}^{(t,k)} := \boldsymbol{\omega}_{\pi_m}^{(t,k)} - \tilde{\boldsymbol{\omega}}_{\pi_m}^{(t,k)}$. By Lemma D.1, for stepsizes $\eta^{(t,k)}$, the one-step update is $(1 + \beta\eta^{(t,k)})$-expansive. As in the convex case, conditioning on whether the touched sample coincides gives

$$\text{w.p. } 1 - \tfrac{1}{MN}: \quad \|\Delta_{\pi_m}^{(t,k+1)}\|_2^2 \leq (1 + \beta\eta^{(t,k)})^2 \|\Delta_{\pi_m}^{(t,k)}\|_2^2, \tag{13}$$

$$\text{w.p. } \tfrac{1}{MN}: \quad \|\Delta_{\pi_m}^{(t,k+1)}\|_2^2 \leq (1 + \beta\eta^{(t,k)})^2 \|\Delta_{\pi_m}^{(t,k)}\|_2^2 + 4(\eta^{(t,k)})^2 \|g_{\pi_m}^{(t,k)}\|_2^2. \tag{14}$$

Taking expectation over the internal randomness,

$$\mathbb{E}\Big[\|\Delta_{\pi_m}^{(t,k+1)}\|_2^2\Big] \leq (1 + \beta\eta^{(t,k)})^2 \mathbb{E}\Big[\|\Delta_{\pi_m}^{(t,k)}\|_2^2\Big] + \frac{4}{MN}(\eta^{(t,k)})^2 \mathbb{E}\Big[\|g_{\pi_m}^{(t,k)}\|_2^2\Big]. \tag{15}$$

**Unrolling over local steps** $k = 0, \ldots, K-1$**.** Define the amplification factors

$$\Phi_t := \prod_{r=0}^{K-1} \big(1 + \beta\eta^{(t,r)}\big)^2, \qquad \Psi_t(k) := \prod_{r=k+1}^{K-1} \big(1 + \beta\eta^{(t,r)}\big)^2 \; (\leq \Phi_t).$$

Iterating equation 49 yields, for each active client $m$,

$$\mathbb{E}\Big[\|\Delta_{\pi_m}^{(t,K)}\|_2^2\Big] \leq \Phi_t \, \mathbb{E}\Big[\|\Delta_{\pi_m}^{(t,0)}\|_2^2\Big] + \frac{4}{MN} \sum_{k=0}^{K-1} (\eta^{(t,k)})^2 \, \Psi_t(k) \, \mathbb{E}\Big[\|g_{\pi_m}^{(t,k)}\|_2^2\Big]. \tag{16}$$

Using $\log(1+x) \leq x$ and $(1+x)^2 \leq e^{2x}$, with $S_t := \sum_{r=0}^{K-1} \eta^{(t,r)}$,

$$\Psi_t(k) \leq \Phi_t = \exp\Big(2 \sum_{r=0}^{K-1} \log(1 + \beta\eta^{(t,r)})\Big) \leq e^{2\beta S_t}. \tag{17}$$

**Sequential pass with partial participation.** Unrolling equation 16 over $m = 1, \ldots, \tilde{M}$ yields a *linear* accumulation of the noise terms (no extra exponential in $\tilde{M}$):

$$\mathbb{E}\Big[\|\Delta^{(t+1)}\|_2^2\Big] = \mathbb{E}\Big[\|\Delta_{\pi_{\tilde{M}}}^{(t,K)}\|_2^2\Big]$$

$$\leq \Phi_t \, \mathbb{E}\big[\|\Delta_T\|_2^2\big] + \frac{4\, e^{2\beta S_t}}{MN} \sum_{m=1}^{\tilde{M}} \sum_{k=0}^{K-1} (\eta^{(t,k)})^2 \, \mathbb{E}\Big[\|g_{\pi_m}^{(t,k)}\|_2^2\Big]$$

$$\leq \Phi_t \, \mathbb{E}\big[\|\Delta_T\|_2^2\big] + \frac{4\, \tilde{M}\, e^{2\beta S_t}}{MN} \sum_{k=0}^{K-1} (\eta^{(t,k)})^2 \, \mathbb{E}\Big[\|g_{\pi_m}^{(t,k)}\|_2^2\Big]. \tag{18}$$

**Averaging round** $T - t_0$ **output.** Assume a burn-in where the two runs coincide at $t_0$, i.e., $\Delta^{(t_0)} = 0$. Averaging equation 18 over $t = t_0, \ldots, T-1$ and using equation 17 gives

$$\mathbb{E}\Big\|\overline{\boldsymbol{\omega}}^{(T)} - \overline{\tilde{\boldsymbol{\omega}}}^{(T)}\Big\|_2^2 = \frac{1}{T - t_0} \sum_{t=t_0}^{T-1} \mathbb{E}\Big[\|\Delta^{(t+1)}\|_2^2\Big] \leq \frac{4\, \tilde{M}}{MN(T-t_0)} \sum_{t=t_0}^{T-1} e^{2\beta S_t} \sum_{k=0}^{K-1} (\eta^{(t,k)})^2 \, \mathbb{E}\Big[\|g_{\pi_m}^{(t,k)}\|_2^2\Big]. \tag{19}$$

By Lemma D.4, for all $(t, k)$, introduce

$$H_t := \sum_{k=0}^{K-1} \big(\eta^{(t,k)}\big)^2, \qquad Q_t := \sum_{k=0}^{K-1} \big(\eta^{(t,k)}\big)^2 \sum_{s=0}^{k} \big(\eta^{(t,s)}\big)^2.$$

Then equation 19 becomes

$$\mathbb{E}\left\|\overline{\boldsymbol{\omega}}^{(T)} - \overline{\widetilde{\boldsymbol{\omega}}}^{(T)}\right\|_2^2 \le \frac{16\,\tilde{M}\,A_\star}{MN(T-t_0)} \sum_{t=t_0}^{T-1} e^{2\beta S_t}\left(H_t + 4\beta^2 Q_t\right). \qquad (20)$$

**Generalization from on-average stability.** By the standard stability-to-generalization conversion (with a tunable $\gamma > 0$),

$$\epsilon_{\text{gen}} = \min_{\gamma, t_0} B(\gamma, t_0) \le \frac{t_0}{MN} + \frac{\beta + \gamma}{2}\,\mathbb{E}\left\|\overline{\boldsymbol{\omega}}^{(T)} - \overline{\widetilde{\boldsymbol{\omega}}}^{(T)}\right\|_2^2 + \frac{3A_\star}{2\gamma}. \qquad (21)$$

We will use $\log(1+x) \le x$, $(1+x)^2 \le e^{2x}$, and $1+x \le e^x$ for $x \ge 0$.

**(i) Square-root schedule** $\eta^{(t,k)} = \dfrac{1}{\sqrt{tK + k + k_0}}$.

By Riemann-sum bounds,

$$S_t = \sum_{k=0}^{K-1} \frac{1}{\sqrt{tK + k + k_0}} \le 2\left(\sqrt{tK + k_0 + K} - \sqrt{tK + k_0}\right) \le \frac{K}{\sqrt{tK + k_0}}, \qquad (22)$$

and

$$H_t = \sum_{k=0}^{K-1} \frac{1}{tK + k + k_0} \le \log\frac{tK + k_0 + K}{tK + k_0}, \qquad Q_t \le H_t^2. \qquad (23)$$

Plugging equation 22–equation 23 into equation 20 gives

$$\mathbb{E}\left\|\overline{\boldsymbol{\omega}}^{(T)} - \overline{\widetilde{\boldsymbol{\omega}}}^{(T)}\right\|_2^2 \le \frac{16\,\tilde{M}\,A_\star}{MN(T-t_0)} \sum_{t=t_0}^{T-1} \exp\left(\frac{2\beta K}{\sqrt{tK+k_0}}\right)\left(\frac{K}{tK+k_0} + 4\beta^2\frac{K^2}{(tK+k_0)^2}\right). \qquad (24)$$

With $u = tK + k_0$ ($du = K\,dt$), $u$ runs from $u_0 := t_0 K + k_0$ to $U := TK + k_0$. Using the change of variables $v = \sqrt{u}$ and $w = \frac{2\beta K}{v}$, we get the tidy bound

$$\mathbb{E}\left\|\overline{\boldsymbol{\omega}}^{(T)} - \overline{\widetilde{\boldsymbol{\omega}}}^{(T)}\right\|_2^2 \lesssim \frac{16\,\tilde{M}\,A_\star}{\beta MN(T-t_0)} \cdot \frac{\sqrt{u_0}}{K} \exp\left(\frac{2\beta K}{\sqrt{u_0}}\right), \qquad u_0 = t_0 K + k_0. \qquad (25)$$

Substitute $t_0 = (TK)^{\frac{2\beta}{1+2\beta}}$ for optimization of $B(\gamma, t_0)$. For large $TK$,

$$\frac{\sqrt{u_0}}{K} \asymp (TK)^{\frac{\beta}{1+2\beta} - \frac{1}{2}}, \qquad \frac{2\beta K}{\sqrt{u_0}} = 2\beta\,(TK)^{\frac{1-\beta}{2(1+2\beta)}}.$$

Hence

$$\mathbb{E}\left\|\overline{\boldsymbol{\omega}}^{(T)} - \overline{\widetilde{\boldsymbol{\omega}}}^{(T)}\right\|_2^2 \lesssim \frac{16\,\tilde{M}\,A_\star}{\beta MNT}\,(TK)^{\frac{\beta}{1+2\beta} - \frac{1}{2}}\,\exp\left(2\beta\,(TK)^{\frac{1-\beta}{2(1+2\beta)}}\right). \qquad (26)$$

Using equation 21 and $T - t_0 \asymp T$, and minimizing $B(\gamma, t_0)$ at $\gamma^\star = \sqrt{3A_\star / \mathbb{E}\|\overline{\boldsymbol{\omega}}^{(T)} - \overline{\widetilde{\boldsymbol{\omega}}}^{(T)}\|_2^2}$, we obtain

$$\boxed{\epsilon_{\text{gen}} \lesssim \frac{(TK)^{\frac{2\beta}{1+2\beta}}}{MN} + \frac{8\,\tilde{M}\,A_\star}{MNT}\,(TK)^{\frac{\beta}{1+2\beta} - \frac{1}{2}}\,e^{2\beta\,(TK)^{\frac{1-\beta}{2(1+2\beta)}}} + 4A_\star\sqrt{\frac{3\tilde{M}}{\beta MNT}\,(TK)^{\frac{\beta}{2(1+2\beta)} - \frac{1}{4}}}\,e^{\beta\,(TK)^{\frac{1-\beta}{2(1+2\beta)}}}.}$$

$$(27)$$

Compared with prior (incorrect) versions, the exponential factor no longer contains $\tilde{M}$; $\tilde{M}$ appears only linearly in the prefactor.

**(ii) Harmonic schedule** $\eta^{(t,k)} = \dfrac{1}{tK + k + k_0}$.

We have

$$S_t = \sum_{k=0}^{K-1} \frac{1}{tK + k + k_0} \le \log \frac{tK + k_0 + K}{tK + k_0}, \qquad H_t = \sum_{k=0}^{K-1} \frac{1}{(tK + k + k_0)^2} \le \frac{1}{tK + k_0}, \qquad (28)$$

$$Q_t \le H_t^2 \le \frac{1}{(tK + k_0)^2}. \qquad (29)$$

Thus

$$e^{2\beta S_t} \le \left( \frac{tK + k_0 + K}{tK + k_0} \right)^{2\beta} \le \exp\left( \frac{2\beta K}{tK + k_0} \right) \le 1 + \frac{2\beta K}{tK + k_0}.$$

Plugging equation 29 into equation 20 and expanding gives

$$\mathbb{E}\left\|\overline{\boldsymbol{\omega}}^{(T)} - \overline{\tilde{\boldsymbol{\omega}}}^{(T)}\right\|_2^2 \le \frac{16\,\tilde{M}\,A_\star}{MN(T - t_0)} \sum_{t=t_0}^{T-1} \left( \frac{1}{tK + k_0} + \frac{2\beta K}{(tK + k_0)^2} + \frac{4\beta^2}{(tK + k_0)^2} + \frac{8\beta^3 K}{(tK + k_0)^3} \right). \qquad (30)$$

With $u = tK + k_0$ and the estimates

$$\sum_{t=t_0}^{T-1} \frac{1}{tK + k_0} \le \frac{1}{K} \log \frac{T}{t_0}, \quad \sum_{t=t_0}^{T-1} \frac{1}{(tK + k_0)^2} \le \frac{1}{K\,u_0}, \quad \sum_{t=t_0}^{T-1} \frac{1}{(tK + k_0)^3} \le \frac{1}{2K\,u_0^2}, \qquad (31)$$

where $u_0 := t_0 K + k_0$, $U := TK + k_0$, we obtain

$$\mathbb{E}\left\|\overline{\boldsymbol{\omega}}^{(T)} - \overline{\tilde{\boldsymbol{\omega}}}^{(T)}\right\|_2^2 \le \frac{16\,\tilde{M}\,A_\star}{MN(T - t_0)} \left[ \frac{1}{K} \log \frac{T}{t_0} + \frac{2\beta}{u_0} + \frac{4\beta^2}{K\,u_0} + \frac{4\beta^3}{u_0^2} \right]. \qquad (32)$$

Substitute $t_0 = (TK)^{\frac{2\beta}{1+2\beta}}$ for optimization. Noting $u_0 \asymp t_0 K = (TK)^{\frac{2\beta}{1+2\beta}} K$ and $T - t_0 \asymp T$, the log term dominates for large $TK$, and we get

$$\mathbb{E}\left\|\overline{\boldsymbol{\omega}}^{(T)} - \overline{\tilde{\boldsymbol{\omega}}}^{(T)}\right\|_2^2 \lesssim \frac{16\,\tilde{M}\,A_\star}{MNTK} \cdot \frac{1}{1 + 2\beta} \log T. \qquad (33)$$

Using equation 21 and $\gamma^\star = \sqrt{3A_\star/\mathcal{S}}$ with $\mathcal{S}$ given by equation 33, we obtain

$$\boxed{\epsilon_{\text{gen}} \lesssim \frac{(TK)^{\frac{2\beta}{1+2\beta}}}{MN} + \frac{8\beta}{1 + 2\beta} \cdot \frac{\tilde{M}\,A_\star \log T}{MNTK} + 4A_\star \sqrt{\frac{3}{1 + 2\beta}} \sqrt{\frac{\tilde{M}\,\log T}{MNTK}}.} \qquad (34)$$

This complete the proof. $\qquad\qquad\square$

### E.2   PROOF OF SFL

#### E.2.1   SETUP.

Let the training set be $S = \{Z_{ij} : i \in [N],\, j \in [M]\}$, with $M$ clients each holding $N$ examples. Let $S^{(ij)}$ denote the neighbour dataset obtained from $S$ by replacing the single example $Z_{ij}$ with an independent copy $\widetilde{Z}_{ij}$. Run the *SFL* algorithm on $S$ and on $S^{(ij)}$ with *identical* internal randomness (same seeds for client participation, permutations, and data indices). Denote the global models after round $t$ by $\boldsymbol{\omega}^{(t)}$ and $\tilde{\boldsymbol{\omega}}^{(t)}$, respectively, with common initialization $\boldsymbol{\omega}^{(0)} = \tilde{\boldsymbol{\omega}}^{(0)}$. Within each round $t$, client $m$ performs $K$ local gradient steps

$$\boldsymbol{\omega}_m^{(t,k+1)} = \boldsymbol{\omega}_m^{(t,k)} - \eta^{(t,k)} \nabla f_{\pi_m}\big(\boldsymbol{\omega}_m^{(t,k)}; Z_{I_m^{(t,k)},m}\big), \qquad k = 0, \ldots, K-1,$$

where $I_m^{(t,k)} \in [N]$ is sampled uniformly (independently across $t, k, m$), and $(\pi_1, \ldots, \pi_M)$ is the client permutation for round $t$. Aggregation is by averaging over the active clients $A_t$ (we write $M_t = |A_t|$); for clarity we present full participation $M_t = M$ (the partial-participation case follows mutatis mutandis with $M_t$). Assume the per-sample loss $f(\cdot; z)$ is $\beta$-smooth; when specified, convexity is also assumed. Define the round-$t$ discrepancy

$$\Delta_t \triangleq \left\|\boldsymbol{\omega}^{(t)} - \tilde{\boldsymbol{\omega}}^{(t)}\right\|_2.$$

### E.2.2 PROOF OF THEOREM 4.5 (SFL ON CONVEX CASE)

**One-step local stability.** Fix $t, k, m$. For brevity write $z = Z_{I_m^{(t,k)}, m}$ and $z' = \widetilde{Z}_{I_m^{(t,k)}, m}$. Because the gradient step map $\Phi_{z,\eta}(x) = x - \eta \nabla f(x; z)$ is 1-expansive for $\eta^{(t,k)} \leq 2/\beta$ (Lemma D.1), we have

$$
\begin{aligned}
\|\boldsymbol{\omega}_m^{(t,k+1)} - \tilde{\boldsymbol{\omega}}_m^{(t,k+1)}\|_2 &= \left\| \Phi_{z, \eta^{(t,k)}}(\boldsymbol{\omega}_m^{(t,k)}) - \Phi_{z', \eta^{(t,k)}}(\tilde{\boldsymbol{\omega}}_m^{(t,k)}) \right\|_2 \\
&\leq \left\| \Phi_{z, \eta^{(t,k)}}(\boldsymbol{\omega}_m^{(t,k)}) - \Phi_{z, \eta^{(t,k)}}(\tilde{\boldsymbol{\omega}}_m^{(t,k)}) \right\|_2 + \eta^{(t,k)} \left\| \nabla f(\tilde{\boldsymbol{\omega}}_m^{(t,k)}; z) - \nabla f(\tilde{\boldsymbol{\omega}}_m^{(t,k)}; z') \right\|_2 \\
&\leq \|\boldsymbol{\omega}_m^{(t,k)} - \tilde{\boldsymbol{\omega}}_m^{(t,k)}\|_2 + \eta^{(t,k)} \left\| \nabla f(\tilde{\boldsymbol{\omega}}_m^{(t,k)}; z) - \nabla f(\tilde{\boldsymbol{\omega}}_m^{(t,k)}; z') \right\|_2. \quad (35)
\end{aligned}
$$

The second term is nonzero only when the sampled pair $(I_m^{(t,k)}, m)$ coincides with the replaced index $(i, j)$. Since the replaced pair $(i, j)$ is uniform over the $MN$ samples, the probability of collision is $1/(MN)$. Using Young's inequality yields for the second term in equation 35:

$$
\mathbb{E}\left[ \|\boldsymbol{\omega}_m^{(t,k+1)} - \tilde{\boldsymbol{\omega}}_m^{(t,k+1)}\|_2^2 \right] \leq \mathbb{E}\left[ \|\boldsymbol{\omega}_m^{(t,k)} - \tilde{\boldsymbol{\omega}}_m^{(t,k)}\|_2^2 \right] + \frac{4(\eta^{(t,k)})^2}{MN} \|g_m^{(t,k)}\|_2^2. \quad (36)
$$

Summing equation 36 over the $K$ local steps and using $\boldsymbol{\omega}_m^{(t,0)} = \boldsymbol{\omega}^{(t)}$, $\tilde{\boldsymbol{\omega}}_m^{(t,0)} = \tilde{\boldsymbol{\omega}}^{(t)}$ gives

$$
\mathbb{E}\left[ \|\boldsymbol{\omega}_m^{(t,K)} - \tilde{\boldsymbol{\omega}}_m^{(t,K)}\|_2^2 \right] \leq \Delta_t^2 + \frac{4}{MN} \sum_{k=0}^{K-1} (\eta^{(t,k)})^2 \|g_{\pi_m}^{(t,k)}\|_2^2. \quad (37)
$$

**Aggregation over clients.** Under partial participation ($\sum_{m=1}^{\tilde{M}} c_m = q \leq 1$), Jensen's inequality gives

$$
\begin{aligned}
\mathbb{E}[\Delta_{t+1}^2] = \mathbb{E}\left\| \sum_{m=1}^{M} c_m (\boldsymbol{\omega}_m^{(t,K)} - \tilde{\boldsymbol{\omega}}_m^{(t,K)}) \right\|_2^2 &\leq \sum_{m=1}^{M} c_m \mathbb{E}\left\| \boldsymbol{\omega}_m^{(t,K)} - \tilde{\boldsymbol{\omega}}_m^{(t,K)} \right\|_2^2 \\
&\leq q\Delta_t^2 + \frac{4q}{MN} \sum_{k=0}^{K-1} (\eta^{(t,k)})^2 \left\| g_m^{(t,k)} \right\|_2^2 \\
&\overset{lem: D.4}{\leq} q\Delta_t^2 + \frac{16q}{MN} \sum_{k=0}^{K-1} (\eta^{(t,k)})^2 \left( 1 + 4\beta^2 \sum_{s=0}^{k} (\eta^{(t,s)})^2 \right) A_\star. \quad (38)
\end{aligned}
$$

**Train outer round $T$.** Introduce the blockwise sums

$$
H_t \triangleq \sum_{k=0}^{K-1} (\eta^{(t,k)})^2, \quad Q_t \triangleq \sum_{k=0}^{K-1} (\eta^{(t,k)})^2 \sum_{s=0}^{k} (\eta^{(t,s)})^2.
$$

Then from equation 38,

$$
\mathbb{E}[\Delta_{t+1}^2] \leq q\Delta_t^2 + \frac{16qA_\star}{MN} H_t + \frac{64q\beta^2 A_\star}{MN} Q_t. \quad (39)
$$

Assume $\Delta_0 = 0$, telescope equation 39 from $t = 0$ to $T - 1$ to get

$$
\boxed{ \mathbb{E}\left\| \overline{\boldsymbol{\omega}}^{(T)} - \overline{\tilde{\boldsymbol{\omega}}}^{(T)} \right\|_2^2 \leq \sum_{t=0}^{T-1} \mathbb{E}[\Delta_t^2] \leq \frac{16qA_\star}{MN(1-q)} \sum_{t=0}^{T-1} H_t + \frac{64q\beta^2 A_\star}{MN(1-q)} \sum_{t=0}^{T-1} Q_t. } \quad (40)
$$

Let $a_k \triangleq (\eta^{(t,k)})^2$. Then

$$
Q_t = \sum_{k=0}^{K-1} a_k \sum_{s=0}^{k} a_s = \sum_{0 \leq s \leq k \leq K-1} a_s a_k = \frac{1}{2}\left( \left( \sum_{k=0}^{K-1} a_k \right)^2 + \sum_{k=0}^{K-1} a_k^2 \right) = \frac{1}{2}\left( H_t^2 + \sum_{k=0}^{K-1} (\eta^{(t,k)})^4 \right),
$$

where we *swap the summation order* to symmetrize the $(s, k)$ pairs.

Consequently,

$$
\frac{1}{2} H_t^2 \leq Q_t \leq H_t^2.
$$

This identity (and bounds) lets us express equation 40 purely in terms of $H_t$ (plus a small quartic correction), which simplifies step-size–specific evaluations.

**Evaluating Two common step-size.** **(i)** For square-root decay step-size $\eta^{(t,k)} = \frac{1}{\sqrt{tK + k + k_0}}$.

$$H_t = \sum_{k=0}^{K-1} \frac{1}{tK + k + k_0} \quad \text{and} \quad \sum_{t=0}^{T-1} H_t = \sum_{n=0}^{TK-1} \frac{1}{n + k_0} \leq \log\left(\frac{TK + k_0 - 1}{k_0 - 1}\right).$$

For each block $t$ we also have the elementary bound:

$$H_t \leq \log\left(\frac{tK + K + k_0 - 1}{tK + k_0 - 1}\right) \leq \frac{K}{tK + k_0 - 1},$$

where we used $\log(1 + x) \leq x$.

Consequently,

$$\sum_{t=0}^{T-1} H_t^2 \leq \sum_{t=0}^{T-1} \frac{K^2}{(tK + k_0 - 1)^2} \leq \frac{K}{k_0 - 1}, \tag{41}$$

the last inequality following from the comparison of the arithmetic progression $tK + k_0 - 1$ with the harmonic tail and the bound $\sum_{n \geq b} n^{-2} \leq 1/(b - 1)$ for $b > 1$.

For the fourth-order terms,

$$\sum_{t=0}^{T-1} \sum_{k=0}^{K-1} (\eta^{(t,k)})^4 = \sum_{t=0}^{T-1} \sum_{k=0}^{K-1} \frac{1}{(tK + k + k_0)^2} = \sum_{n=0}^{TK-1} \frac{1}{(n + k_0)^2} \leq \frac{1}{k_0 - 1}.$$

Then we get:

$$\sum_{t=0}^{T-1} Q_t \leq \frac{1}{2}\left(\sum_{t=0}^{T-1} H_t^2 + \sum_{t=0}^{T-1} \sum_{k=0}^{K-1} (\eta^{(t,k)})^4\right) \leq \frac{K + 1}{2(k_0 - 1)}. \tag{42}$$

Plugging equation 41 and equation 42 into equation 40 yields the explicit squared stability bound for the square-root schedule:

$$\boxed{\mathbb{E}[\Delta_T^2] \leq \frac{16qA_\star}{MN(1 - q)}\left(\log\left(\frac{TK + k_0 - 1}{k_0 - 1}\right) + \frac{2\beta^2(K + 1)}{(k_0 - 1)}\right).} \tag{43}$$

**(ii)** For harmonic decay step-size $\eta^{(t,k)} = \frac{1}{tK + k + k_0}$. Now $\left(\eta^{(t,k)}\right)^2 = \frac{1}{(tK + k + k_0)^2}$.

$$H_t = \sum_{k=0}^{K-1} \frac{1}{(tK + k + k_0)^2} \leq \int_{tK + k_0 - 1}^{tK + K + k_0 - 1} \frac{dx}{x^2} = \frac{1}{tK + k_0 - 1} - \frac{1}{tK + K + k_0 - 1}.$$

Summing $t = 0, \ldots, T - 1$ yields

$$\sum_{t=0}^{T-1} H_t \leq \frac{1}{k_0 - 1} - \frac{1}{TK + k_0 - 1} \leq \frac{1}{k_0 - 1}. \tag{44}$$

$$\sum_{t=0}^{T-1} H_t^2 \leq \sum_{t=0}^{T-1} \frac{1}{(tK + k_0 - 1)^2} \leq \frac{1}{K(k_0 - 1)},$$

$$\sum_{t=0}^{T-1} \sum_{k=0}^{K-1} (\eta^{(t,k)})^4 = \sum_{n=0}^{TK-1} \frac{1}{(n + k_0)^4} \leq \sum_{n=k_0}^{\infty} \frac{1}{n^4} \leq \frac{1}{3(k_0 - 1)^3},$$

Therefore,

$$\sum_{t=0}^{T-1} Q_t \leq \frac{1}{2}\left(\frac{1}{K(k_0 - 1)} + \frac{1}{3(k_0 - 1)^3}\right). \tag{45}$$

Substituting equation 44 and equation 45 into equation 40 yields the explicit squared stability bound for the harmonic schedule:

$$\boxed{\mathbb{E}[\Delta_T^2] \leq \frac{16qA_\star}{MN(1 - q)}\left(\frac{1}{k_0 - 1} + \frac{2\beta^2}{K(k_0 - 1)} + \frac{2\beta^2}{3(k_0 - 1)^3}\right).} \tag{46}$$

**From stability to generalization.** By Theorem **??**, to obtain the tightest bound, minimize the right-hand side with respect to $\gamma > 0$. The generalization bound is then

$$\min_\gamma B(\gamma) = \frac{3A_\star}{2\gamma} + \frac{\beta + \gamma}{2}\,\mathbb{E}\big[\Delta_T^2\big], \qquad \gamma^* = \sqrt{\frac{3A_\star}{\mathbb{E}\big[\Delta_T^2\big]}}.$$

**(i)** For square-root decay, using equation 43, the generalization bound is

$$\epsilon_{\mathrm{gen}} \le 4\sqrt{3}\,A_\star \sqrt{\frac{q}{MN(1-q)}\left(\log\frac{TK+k_0-1}{k_0-1} + \frac{2\beta^2(K+1)}{k_0-1}\right)} + \frac{8\beta\,q\,A_\star}{MN(1-q)}\left(\log\frac{TK+k_0-1}{k_0-1} + \frac{2\beta^2(K+1)}{k_0-1}\right).$$

For large $T, K, M$ and constant $k_0$:

$$\epsilon_{\mathrm{gen}} \le \tilde{O}\left(\sqrt{\frac{q\,(\log(TK) + \beta^2 K)}{MN(1-q)}} + \frac{\beta\,q\,(\log(TK) + \beta^2 K)}{MN(1-q)}\right).$$

**(ii)** For harmonic decay, using equation 46, the generalization bound is

$$\epsilon_{\mathrm{gen}} \le 4\sqrt{3}\,A_\star \sqrt{\frac{q}{MN(1-q)}\left(\frac{1}{k_0-1} + \frac{2\beta^2}{K(k_0-1)} + \frac{2\beta^2}{3(k_0-1)^3}\right)} + \frac{8\beta\,q\,A_\star}{MN(1-q)}\left(\frac{1}{k_0-1} + \frac{2\beta^2}{K(k_0-1)} + \frac{2\beta^2}{3(k_0-1)^3}\right).$$

For large $K$ and constant $k_0$:

$$\epsilon_{\mathrm{gen}} \le \tilde{O}\left(\sqrt{\frac{q}{NM(1-q)}} + \frac{\beta\,q}{NM(1-q)}\right).$$

This completes the proof. $\qquad\square$

### E.2.3  Proof of Theorem 4.6 (SFL On Non-convex Case)

**One-step local stability after the burn-in index** $t_0$. Write $\Delta_m^{(t,k)} := \boldsymbol{\omega}_m^{(t,k)} - \tilde{\boldsymbol{\omega}}_m^{(t,k)}$. Under Assumption 4.1 and Lemma D.1, the one-step update is $(1 + \beta\eta^{(t,k)})$-expansive. Conditioning on whether the touched sample coincides:

$$\text{w.p. } 1 - \tfrac{1}{MN}: \quad \|\Delta_m^{(t,k+1)}\|_2^2 \le (1 + \beta\eta^{(t,k)})^2 \|\Delta_m^{(t,k)}\|_2^2, \tag{47}$$

$$\text{w.p. } \tfrac{1}{MN}: \quad \|\Delta_m^{(t,k+1)}\|_2^2 \le (1 + \beta\eta^{(t,k)})^2 \|\Delta_m^{(t,k)}\|_2^2 + 4(\eta^{(t,k)})^2 \|g_m^{(t,k)}\|_2^2. \tag{48}$$

Taking expectation:

$$\mathbb{E}\Big[\|\Delta_m^{(t,k+1)}\|_2^2\Big] \le (1 + \beta\eta^{(t,k)})^2 \mathbb{E}\Big[\|\Delta_m^{(t,k)}\|_2^2\Big] + \frac{4}{MN}(\eta^{(t,k)})^2\,\mathbb{E}\Big[\|g_m^{(t,k)}\|_2^2\Big]. \tag{49}$$

**Unrolling over local steps** $k = 0, \ldots, K-1$. Define

$$\Phi_t := \prod_{r=0}^{K-1}\big(1 + \beta\eta^{(t,r)}\big)^2, \qquad \Psi_t(k) := \prod_{r=k+1}^{K-1}\big(1 + \beta\eta^{(t,r)}\big)^2 \;(\le \Phi_t).$$

Iterating equation 49 yields

$$\mathbb{E}\Big[\|\Delta_m^{(t,K)}\|_2^2\Big] \le \Phi_t\,\mathbb{E}\Big[\|\Delta_m^{(t,0)}\|_2^2\Big] + \frac{4}{MN}\sum_{k=0}^{K-1}(\eta^{(t,k)})^2\,\Psi_t(k)\,\mathbb{E}\Big[\|g_{\pi_m}^{(t,k)}\|_2^2\Big]. \tag{50}$$

With $S_t := \sum_{k=0}^{K-1}\eta^{(t,k)}$ and using $\log(1+x) \le x$, $(1+x)^2 \le e^{2x}$,

$$\Psi_t(k) \le \Phi_t = \exp\!\Big(2\sum_{r=0}^{K-1}\log(1+\beta\eta^{(t,r)})\Big) \le \exp\!\Big(2\sum_{r=0}^{K-1}\beta\eta^{(t,r)}\Big) = e^{2\beta S_t}. \tag{51}$$

**Aggregation over active clients.** Let the server aggregate by weighted averaging $\bar{\boldsymbol{\omega}}^{(t,K)} := \sum_{m \in \mathcal{M}_t} c_m \boldsymbol{\omega}_m^{(t,K)}$ with $c_m \geq 0$, $\sum_{m \in \mathcal{M}_t} c_m = q \leq 1$, and $\boldsymbol{\omega}^{(t+1)} := \bar{\boldsymbol{\omega}}^{(t,K)}$. By Jensen's inequality and equation 50,

$$
\begin{aligned}
\mathbb{E}\big[\|\Delta^{(t+1)}\|_2^2\big] &= \mathbb{E}\bigg[\Big\|\sum_{m \in \mathcal{M}_t} c_m\big(\boldsymbol{\omega}_m^{(t,K)} - \tilde{\boldsymbol{\omega}}_m^{(t,K)}\big)\Big\|_2^2\bigg] \\
&\leq \sum_{m \in \mathcal{M}_t} c_m \, \mathbb{E}\big[\|\Delta_m^{(t,K)}\|_2^2\big] \\
&\leq q\Phi_t \, \mathbb{E}\big[\|\Delta_t\|_2^2\big] + \frac{4q}{MN} \sum_{k=0}^{K-1} (\eta^{(t,k)})^2 \, \Psi_t(k) \, \mathbb{E}\big[\|g_{\pi_m}^{(t,k)}\|_2^2\big].
\end{aligned}
\tag{52}
$$

Accumulating from $t_0$ (with $\Delta^{(t_0)} = 0$) to $T - 1$ and summing the geometric factor coming from $q\Phi_t$ gives

$$
\begin{aligned}
\mathbb{E}\big[\|\Delta_T\|_2^2\big] &= \sum_{t=t_0}^{T-1} \mathbb{E}\big[\|\Delta^{(t+1)}\|_2^2\big] \\
&\leq \frac{4q}{MN(1-q)} \sum_{t=t_0}^{T-1} e^{2\beta S_t} \sum_{k=0}^{K-1} (\eta^{(t,k)})^2 \, \mathbb{E}\big[\|g_{\pi_m}^{(t,k)}\|_2^2\big].
\end{aligned}
\tag{53}
$$

By Lemma D.4, define $H_t := \sum_{k=0}^{K-1} \big(\eta^{(t,k)}\big)^2$, $\quad Q_t := \sum_{k=0}^{K-1} \big(\eta^{(t,k)}\big)^2 \sum_{s=0}^{k} \big(\eta^{(t,s)}\big)^2$. Then from equation 53,

$$
\boxed{\mathbb{E}\big[\|\Delta_T\|_2^2\big] \leq \frac{16qA_\star}{MN(1-q)} \sum_{t=t_0}^{T-1} e^{2\beta S_t} \Big(H_t + 4\beta^2 Q_t\Big).}
\tag{54}
$$

**Generalization via stability.** For any $\gamma > 0$ and burn-in $t_0$,

$$
\epsilon_{\mathrm{gen}} = \min_{\gamma, t_0} \Big\{ \frac{t_0}{MN} + \frac{\beta + \gamma}{2} \mathbb{E}\big[\|\Delta_T\|_2^2\big] + \frac{3A_\star}{2\gamma} \Big\}.
\tag{55}
$$

It remains to bound $S_t, H_t, Q_t$.

**(i) Square-root schedule** $\eta^{(t,k)} = \dfrac{1}{\sqrt{tK + k + k_0}}$. Using $S_t \leq \frac{K}{\sqrt{tK+k_0}}$, $H_t \leq \frac{K}{tK+k_0}$, $Q_t \leq \frac{K^2}{(tK+k_0)^2}$, equation 54 becomes

$$
\mathbb{E}\big[\|\Delta_T\|_2^2\big] \leq \frac{16qA_\star}{MN(1-q)} \sum_{t=t_0}^{T-1} \exp\Big(\frac{2\beta K}{\sqrt{tK+k_0}}\Big)\Big(\frac{K}{tK+k_0} + 4\beta^2 \frac{K^2}{(tK+k_0)^2}\Big).
\tag{56}
$$

Let $u = tK + k_0$ ($du = K\,dt$) and $v = \sqrt{u}$. As in the SSL analysis, one obtains

$$
\int_{u_0}^{U} \exp\Big(\frac{2\beta K}{\sqrt{u}}\Big)\Big(\frac{K}{u} + 4\beta^2 \frac{K^2}{u^2}\Big) du \lesssim \frac{2K\sqrt{u_0}}{\beta} e^{\frac{2\beta K}{\sqrt{u_0}}}, \qquad u_0 := t_0 K + k_0, \; U := TK + k_0.
\tag{57}
$$

Therefore

$$
\mathbb{E}\big[\|\Delta_T\|_2^2\big] \lesssim \frac{16qA_\star}{\beta\, MN(1-q)} \sqrt{u_0} \exp\Big(\frac{2\beta K}{\sqrt{u_0}}\Big).
\tag{58}
$$

Choosing $t_0 = (TK)^{\frac{2\beta}{1+2\beta}}$ gives

$$
\sqrt{u_0} \asymp (TK)^{\frac{\beta}{1+2\beta}} K^{1/2}, \qquad \frac{2\beta K}{\sqrt{u_0}} = 2\beta\,(TK)^{\frac{1-\beta}{2(1+2\beta)}}.
$$

Hence

$$\mathbb{E}\big[\|\Delta_T\|_2^2\big] \;\lesssim\; \frac{16qA_\star}{\beta\,MN(1-q)}\,(TK)^{\frac{\beta}{1+2\beta}}\,K^{\frac{1}{2}}\,\exp\!\Big(2\beta\,(TK)^{\frac{1-\beta}{2(1+2\beta)}}\Big).\qquad(59)$$

Using equation 55 with $\gamma^\star = \sqrt{3A_\star/\mathbb{E}\|\Delta_T\|^2}$,

$$\boxed{\epsilon_{\text{gen}} \;\lesssim\; \frac{(TK)^{\frac{2\beta}{1+2\beta}}}{MN} + \frac{8qA_\star}{MN(1-q)}\,(TK)^{\frac{\beta}{1+2\beta}}\,K^{\frac{1}{2}}\,e^{2\beta\,(TK)^{\frac{1-\beta}{2(1+2\beta)}}} \;+\; 4\sqrt{3}\,A_\star\sqrt{\frac{q}{\beta\,MN(1-q)}}\,(TK)^{\frac{\beta}{2(1+2\beta)}}\,K^{\frac{1}{4}}\,e^{\beta\,(TK)^{\frac{1-\beta}{2(1+2\beta)}}}.}$$
$$(60)$$

**(ii) Harmonic schedule** $\eta^{(t,k)} = \frac{1}{tK+k+k_0}$. Here $S_t \le \log\big(\frac{tK+k_0+K}{tK+k_0}\big)$, $H_t \le \frac{1}{tK+k_0}$, $Q_t \le \frac{1}{(tK+k_0)^2}$, and $e^{2\beta S_t} \le \exp\big(\frac{2\beta K}{tK+k_0}\big) \le 1 + \frac{2\beta K}{tK+k_0}$. Summing by integral comparison gives, with $u_0 := t_0 K + k_0$,

$$\sum_{t=t_0}^{T-1}\frac{1}{tK+k_0} \le \frac{1}{K}\log\frac{T}{t_0}, \quad \sum_{t=t_0}^{T-1}\frac{1}{(tK+k_0)^2} \le \frac{1}{K\,u_0}, \quad \sum_{t=t_0}^{T-1}\frac{1}{(tK+k_0)^3} \le \frac{1}{2K\,u_0^2}.\qquad(61)$$

Thus

$$\mathbb{E}\big[\|\Delta_T\|_2^2\big] \;\le\; \frac{4qA_\star}{MN(1-q)}\left[\frac{1}{K}\log\frac{T}{t_0} + \frac{2\beta}{u_0} + \frac{4\beta^2}{K\,u_0} + \frac{4\beta^3}{u_0^2}\right].\qquad(62)$$

Choose $t_0 = (TK)^{\frac{2\beta}{1+2\beta}}$, so that the log term dominates:

$$\mathbb{E}\big[\|\Delta_T\|_2^2\big] \;\lesssim\; \frac{4qA_\star}{MN(1-q)} \cdot \frac{1}{K} \cdot \frac{1}{1+2\beta}\,\log T.\qquad(63)$$

Plugging equation 63 into equation 55 with $\gamma^\star$ yields

$$\boxed{\epsilon_{\text{gen}} \;\lesssim\; \frac{(TK)^{\frac{2\beta}{1+2\beta}}}{MN} + \frac{2\beta}{1+2\beta} \cdot \frac{qA_\star \log T}{MN(1-q)\,K} + 2A_\star\sqrt{\frac{3q}{(1+2\beta)\,MN(1-q)\,K}}\,\sqrt{\log T}.}\qquad(64)$$

This complete the proof. $\qquad\square$