# OpenReview forum: "Stability and Generalization of Split Learning : Sequential or Federated"
_ICLR.cc/2026/Conference — ICLR 2026 Conference Withdrawn Submission_

### Official Review · Reviewer_grj2 · 2025-10-24

**Soundness:** 3
**Presentation:** 3
**Contribution:** 1
**Rating:** 2
**Confidence:** 4

**Summary:**

This paper provides a theoretical analysis of the generalization error for Split Learning. The work focuses on two prominent paradigms: Sequential Split Learning and Split Federated Learning. The authors derive generalization bounds for both algorithms under convex and non-convex settings, aiming to provide insights into their stability and performance.

**Strengths:**

1.The primary contribution of this paper is that it presents the first theoretical framework for analyzing the generalization error of SSL and SFL. Establishing a theoretical foundation for the generalization properties of these methods is a valuable research direction.

**Weaknesses:**

1.Insufficient Comparative Analysis and Practical Guidance: While the paper's core contribution is providing the first generalization analysis for SSL and SFL, it lacks a direct comparative analysis between the two. The paper simply lists the generalization upper bounds for SSL and SFL without offering clear guidance on which method is preferable under different scenarios.
2.Limited Novelty: The novelty of the work is questionable, as there are existing analyses on the convergence bounds of SSL and SFL[1]. Generalization is a closely related field to convergence, and a more thorough discussion of how this work's theoretical contributions significantly advance the state of the art beyond existing convergence results is needed.
3.Contradiction Between Theory and Experiments: The theoretical analysis concludes that the stability and generalization bounds improve as the number of participating clients per round decreases (i.e., a smaller M̃ for SSL or a smaller participation rate q for SFL). This directly contradicts the empirical observation in the experimental section, which states that "higher client participation consistently improves stability." The authors do not provide any explanation for this significant discrepancy.
4.Insufficient Experiments: The experiments are not comprehensive enough to fully validate the theoretical claims. A more robust evaluation would involve a wider range of datasets, models, and more detailed ablation studies to investigate the impact of key factors identified in the theory, such as data heterogeneity and the number of local steps.
5.Formatting Issues: There are some formatting problems that need correction. For instance, in Theorem 4.5, the equation extends beyond the page margin, which affects readability.

[1] Li, Yipeng, and Xinchen Lyu. "Convergence analysis of sequential federated learning on heterogeneous data." Advances in Neural Information Processing Systems 36 (2023): 56700-56755.

**Questions:**

1.Can the authors elaborate on the practical implications of their theoretical bounds? Based on the derived bounds, could you provide a clear comparative analysis of when SSL should be preferred over SFL, and vice versa?
2.How do the authors reconcile the contradiction between the theoretical finding that fewer participating clients lead to a better theoretical bound and the experimental result that higher participation improves stability? Does this point to a gap in the theoretical model or a specific condition in the experimental setup?
3.Could the authors clarify the key technical innovations of their analysis in comparison to the existing convergence analyses for SSL and SFL?
4.The experiments show that higher client participation is always better. Could the authors design an experiment where, as the theory suggests, lower client participation actually leads to better generalization performance to validate the tightness of the bound?

---

### Official Review · Reviewer_itNX · 2025-10-27

**Soundness:** 3
**Presentation:** 3
**Contribution:** 2
**Rating:** 4
**Confidence:** 4

**Summary:**

This paper provides a first generalization analysis for split learning. It focuses on two primary schemes, namely sequential split learning (SSL) and split federated learning (SFL, v1) and analyzes both convex and non-convex objectives with varying learning rate strategies. The whole proof is based off an l_2 model stability framework. Some experiments on MNIST and CIFAR-10 are provided to evaluate the performance under various system parameters. The paper is well organized and generally easy to read.

**Strengths:**

- To the reviewer's best knowledge, this is the first work to analyze generalization error of split learning, which is an important contribution to the distributed learning community. The analysis is rather comprehensive covering convex and non-convex cases with different learning rate strategies. The reviewer gave a quick read over the proofs and no major flaws were detected.

- Some interesting discussions were provided regarding the choice of SSL and SFL.

- The paper is generally well written and easy to follow.

**Weaknesses:**

The reviewer has severalmajor concerns regarding the current version:

- When looking at the proofs, the entire framework is largely based on the previous generalization analysis on federated learning.  In fact, the proofs inaccurately used the global model definition w to represent client-side model which should have been something like w_c. Since the  separation between client-side and server-side models in SL is a key difference to FL, hence without detailing this in the proofs it would be difficult to assess the technical challenges/contributions of the paper.

- The authors stated in Appendix B that SFL-V2 would achieve similar results to SFL-V1 without providing proofs, which the reviewer found to be not convincing. SFL-v2 requires the server-side model to sequentially interact with the client-side models, while SFL-V1 operates essentially the same to FL. Hence, the reviewer is expecting different results between the two.

- Another major concern is that all generalization errors converge to zero with large M or N, even under non-convex and data heterogeneity. This basically implies that the client drift issue disappears with a large client number, which is surprising. Instead, the reviewer is expecting an error term related to data heterogeneity that does not diminish to zero with large M or N. Can the authors clarify on this?

There are also some other minor concerns:

- In Figs. 3-4, the vertical axis is a stability measure, but what does x'_t mean? Also, should x be w? In addition, the reviewer expects the measure to be a difference between train and test errors. Can the authors clarify on this?

- Some typos are scattered, e.g., in line 143, "generakization" -> "generalization"; in Fig. 3e, after "size" there should be "on"; in line 1404, the theorem numbering is missing.

**Questions:**

See above.

---

### Official Review · Reviewer_nHte · 2025-10-31

**Soundness:** 3
**Presentation:** 2
**Contribution:** 2
**Rating:** 4
**Confidence:** 3

**Summary:**

The paper presents the first theoretical framework for analyzing the generalization error of Split Learning algorithms, specifically focusing on Sequential Split Learning and Split Federated Learning. The authors use an on-average stability approach to account for client drift and biased gradient estimates, which arise from local updates and data aggregation.

The paper introduces a new framework that connects the optimization process and generalization in SL. This framework provides insights into how SSL and SFL differ in their stability and generalization behaviors. The authors demonstrate that SSL is particularly effective in scenarios with sparse client participation and long-horizon training. In contrast, SFL shows advantages in non-convex settings with balanced client participation.

The paper derives precise stability bounds for both convex and non-convex settings. The theoretical predictions are validated through experiments on MNIST and CIFAR-10 benchmarks. The experimental results align with the theoretical findings.

**Strengths:**

The paper provides a theoretical analysis of the generalization error in Split Learning. While convergence analyses for SSL and SFL have been established, their generalization bounds were unexplored before.

The use of an on-average stability approach allows for a more detailed analysis of the impact of local update drift and aggregation-induced errors. The distinction between the optimal use cases for SSL and SFL, backed by theoretical bounds and experimental results is a valuable practical contribution.

**Weaknesses:**

I believe the derived theoretical bounds are complex which makes them less clear to a broad audience and could pose challenges for practical application without further simplification or interpretation.

The experiments use relatively small-scale datasets like MNIST and CIFAR-10. They provide initial validation, but the paper could be strengthened by testing the framework on a wider range of larger and more complex datasets and models, particularly in the context of LLMs, which are mentioned as a key application area.

While the application to Split Learning is new, the core technique of using on-average stability for generalization analysis is not. The paper builds upon existing work on stability and generalization in other machine learning paradigms.

For deep neural networks, especially modern architectures, smoothness can vary across the loss landscape. Some regions might be relatively flat (small $\beta$), while others, can be non-smooth (large or even undefined $\beta$). The $\beta$-smoothness assumption is mathematically convenient, but in practice $\beta$ is not fixed. Also, it is architecture dependent.

Bounded Heterogeneity assumption in reality rarely holds and is highly restrictive.  In many realistic SL scenarios, heterogeneity can be extreme. Consider a federated keyboard model where some users primarily type in slang while others use formal language. In these cases, the local gradients can point in nearly opposite directions. The assumption that holds for all model weights is highly restrictive. As the model trains, it might enter regions where client gradients diverge more significantly.

The conclusion that SFL excels in heterogeneous non-convex systems is relying on this bounded heterogeneity assumption.

**Questions:**

Do you see challenges with the $\beta$-smoothness assumption in the loss landscapes of Transformers? Would the theoretical guidance on choosing between SSL and SFL still hold in an LLM training scenario?

How would a system designer ever know the value of $\zeta^2?$

Beyond the guidance of more clients is better or harmonic decay is robust, what is the advice your theory provides to a practitioner looking to tune their SL system for better generalization?

Could a server create a drift-correction term that is applied during aggregation? Could you move from an analysis that accepts drift as a given to an algorithm that treats it as a controllable variable?

---

### Official Review · Reviewer_kiou · 2025-11-04

**Soundness:** 2
**Presentation:** 2
**Contribution:** 2
**Rating:** 4
**Confidence:** 3

**Summary:**

The paper studies the stability and generalization of sequential and federated split learning (SSL and SFL). Leveraging on-average stability, the paper theoretically justifies generalization performances of SSL and SFL related to different hyperparameter settings. Finally theoretical results are validated by numerical experiments.

**Strengths:**

The paper proposes the first theoretical analysis of generalization of SSL and SFL, quantify their advantages under different settings.

**Weaknesses:**

Since on-average stability applied to centralized and federated learning for generalization analysis is well-studied, could the authors explain what the main difficulty and technical contribution of their analysis and results?

**Questions:**

See weakness.

---

### Note · Authors · 2025-11-23

I have read and agree with the venue's withdrawal policy on behalf of myself and my co-authors.